# Innexin function dictates the spatial relationship between distal somatic cells in the *Caenorhabditis elegans* gonad without impacting the germline stem cell pool

Theadora Tolkin[1,2†], Ariz Mohammad[3†], Todd A Starich[4], Ken CQ Nguyen[5], David H Hall[5], Tim Schedl[3*‡], E Jane Albert Hubbard[1,2,6*‡], David Greenstein[4*‡]

[1]Kimmel Center for Biology and Medicine at the Skirball Institute, NYU Grossman School of Medicine, New York, United States; [2]Department of Cell Biology, NYU Grossman School of Medicine, New York, United States; [3]Department of Genetics, Washington University School of Medicine, St. Louis, United States; [4]Department of Genetics, Cell Biology and Development, University of Minnesota, Minneapolis, United States; [5]Department of Neuroscience, Albert Einstein College of Medicine, The Bronx, United States; [6]Department of Pathology, NYU Grossman School of Medicine, New York, United States

*For correspondence:
ts@wustl.edu (TS);
jane.hubbard@med.nyu.edu
(EJAH);
green959@umn.edu (DG)

†These authors contributed
equally to this work
‡These authors also contributed
equally to this work

Competing interest: See page
22

Reviewing Editor: Yukiko M
Yamashita, Whitehead Institute/
MIT, United States

**Abstract** Gap-junctional signaling mediates myriad cellular interactions in metazoans. Yet, how gap junctions control the positioning of cells in organs is not well understood. Innexins compose gap junctions in invertebrates and affect organ architecture. Here, we investigate the roles of gap-junctions in controlling distal somatic gonad architecture and its relationship to underlying germline stem cells in *Caenorhabditis elegans*. We show that a reduction of soma–germline gap-junctional activity causes displacement of distal sheath cells (Sh1) towards the distal end of the gonad. We confirm, by live imaging, transmission electron microscopy, and antibody staining, that bare regions—lacking somatic gonadal cell coverage of germ cells—are present between the distal tip cell (DTC) and Sh1, and we show that an innexin fusion protein used in a prior study encodes an antimorphic gap junction subunit that mispositions Sh1. We determine that, contrary to the model put forth in the prior study based on this fusion protein, Sh1 mispositioning does not markedly alter the position of the borders of the stem cell pool nor of the progenitor cell pool. Together, these results demonstrate that gap junctions can control the position of Sh1, but that Sh1 position is neither relevant for GLP-1/Notch signaling nor for the exit of germ cells from the stem cell pool.

## Editor's evaluation

This manuscript is important and of interest to a broad audience of stem cell scientists interested in the regulation of stem cell niche architecture. The authors provide a detailed description of the effects of the loss of innexin, a gap junction protein, on the somatic cells that support the distal, proliferative end of the *C. elegans* germ line. The key findings of this manuscript pose a strong challenge to a recent new model of somatic sheath Sh1 cells determining the boundary of germline stem cell niche compartment. Discrepancies remain between the current manuscript and the manuscript by Li et al., which must be resolved in the future.

## Introduction

The relative positions of certain cells within larger organ structures are often important for tissue function. Yet the mechanisms by which cells reach and maintain their precise relative positions within organs are poorly defined. Gap junctions act as conduits for small molecules passed between cells and/or as rivets to ensure adhesion between cells (reviewed by *Skerrett and Williams, 2017*). They have also been implicated in cell morphology within organs (reviewed by *Phelan, 2005*), however this latter role is less well characterized. Here, we take advantage of well-characterized and stereotypical morphology, interactions, and relationships among cells in *Caenorhabditis elegans* to investigate the role gap junctions play in somatic gonad architecture and its consequences, or lack thereof, for the underlying germline stem cells.

The *C. elegans* hermaphrodite gonad provides a premier system for studying organogenesis and stem cell behavior (reviewed by *Hubbard and Greenstein, 2000*; *Hubbard and Schedl, 2019*). Two gonad arms, anterior or posterior of a central uterus and vulva, are each capped by a single somatic cell, the distal tip cell (DTC), that establishes a stem cell niche. Germline stem cells and their proliferative progeny, which together are referred to as progenitors, are maintained by GLP-1/Notch-mediated signaling in the germ line in response to DSL family ligands LAG-2 and APX-1 produced by the DTC (*Austin and Kimble, 1987*; *Berry et al., 1997*; *Henderson et al., 1994*; *Nadarajan et al., 2009*; *Yochem and Greenwald, 1989*). Proximal to the DTC, five pairs of sheath cells (named as pairs Sh1 to Sh5, distal to proximal) provide additional support to the germ line. In particular, Sh1 is implicated in promoting germline progenitor cell proliferation (*Killian and Hubbard, 2005*; *McCarter et al., 1997*). Although the molecular and cellular mechanisms by which Sh1 promotes germline proliferation remain to be fully elucidated, it is clear that one mechanism for the function of these cells involves the formation of gap junctions with germ cells (*Starich et al., 2014*).

Invertebrate gap junctions are formed from octameric hemichannels of innexin proteins (*Oshima et al., 2016*). In *C. elegans*, INX-8 and INX-9 associate to form hemichannels in the hermaphrodite somatic gonad which couple to germline innexin hemichannels (INX-14 with INX-21 or INX-22) to promote germline proliferation and inhibit meiotic maturation, respectively (*Figure 1A–B*; *Starich et al., 2014*). Phenotypic analysis of reduction-of-function mutants in *inx-8* recently led to the discovery of malonyl-CoA as a key cargo that traverses the soma-germline junctions to ensure proper embryogenesis (*Starich et al., 2020*).

In the distal gonad, the somatic gonadal hemichannel components INX-8 and INX-9 are required redundantly for germ cell proliferation and differentiation. Loss of both components renders the germ line devoid of all but a handful of germ cells, which fail to undergo gametogenesis. Restoration of *inx-8* either to the DTC or to sheath cells rescues the severe germline proliferation defect of the *inx-8(0) inx-9(0)* double mutant (*Starich et al., 2014*). In addition, a reduction of *inx-8* and *inx-9* via hypomorphic alleles or by RNA interference limits expansion of the pool of proliferative germ cells (*Dalfó et al., 2020*; *Starich and Greenstein, 2020*; *Starich et al., 2014*). The mutant innexin allele combinations used in this study are proficient in promoting germline proliferation; however, as this work shows, many of them affect the position of Sh1 with respect to the distal end of the gonad.

Several prior observations point to a role for innexins in overall somatic gonad architecture. In young adult hermaphrodites, the DTC forms long extending processes reaching proximally towards Sh1, while the distal border of Sh1 is more regular, with filopodia extending distally towards the DTC. Extensive ultrastructural and both fixed and live image analysis demonstrated the existence of 'bare regions' in between the proximal-extending DTC processes and the distal-extending filopodial extensions of Sh1 in the adult hermaphrodite gonad. In these regions, germ cells are in contact with neither the DTC nor Sh1, but are covered only by a basal lamina (*Hall et al., 1999*). Interestingly, the hypomorphic allele *inx-14(ag17)* (*Miyata et al., 2008*) causes Sh1 to reach almost to the distal end of the gonad (*Starich et al., 2014*) and to obliterate any bare region between the DTC and Sh1. Similarly, a loss of bare regions was observed in *inx-8(0) inx-9(0)* double mutants in which germline proliferation was largely restored through expression of an *inx-8::gfp* transgene in the DTC only (*Starich et al., 2014*). This latter result suggested that if Sh1 cannot form gap junctions with germ cells, Sh1 becomes distally mispositioned. However, the consequences of this mispositioning and the accompanying loss of the bare regions for germline stem cells have not been previously explored.

The spatial relationship between the DTC, Sh1 and the germ line summarized above was recently challenged, and an hypothesis put forth that Sh1 might guide an oriented and asymmetric division

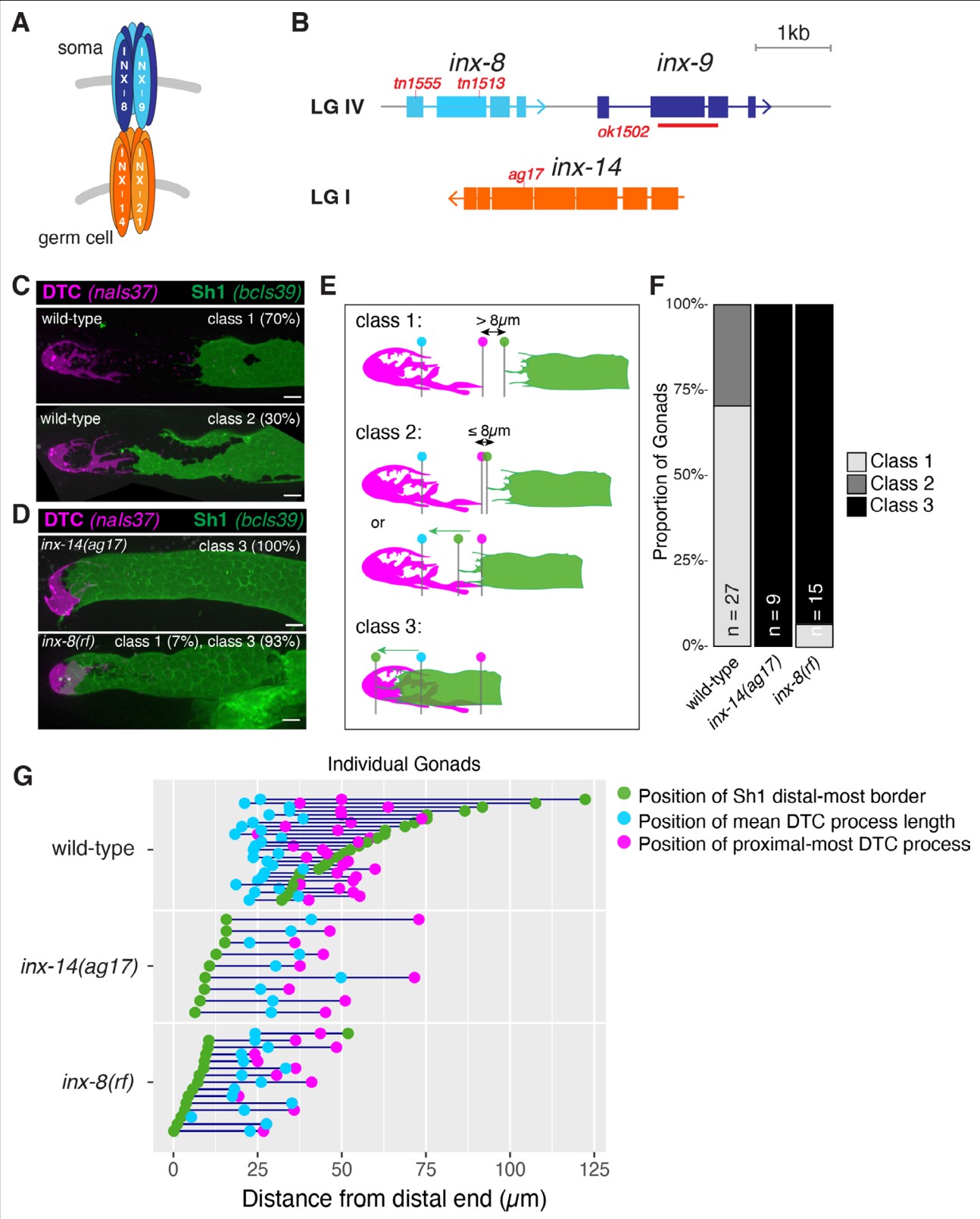

**Figure 1.** Germline and somatic gonad innexins are required for proper somatic gonad architecture. (**A**) Schematic of paired somatic and germline octameric hemichannels. (**B**) Schematic diagram of the *inx-8 inx-9* locus and the *inx-14* locus, with relevant mutations indicated in red. (**C**) Fluorescent confocal maximum projection images of distal gonads in live worms. Distal tip cell (DTC) marked in magenta (*naIs37[lag-2p::mCherry-PH]*) and sheath pair 1 (Sh1) marked in green (*bcIs39[lim-7p::CED-1::GFP]*). Top: strain bearing markers only, denoted "wild-type", representative of phenotypic Classes

*Figure 1 continued on next page*

*Figure 1 continued*

1 and 2. (**D**) *inx-14(ag17)* and *inx-8(tn1513tn1555) inx-9(ok1502)*, denoted '*inx-8(rf)*' after *Starich et al., 2020*, representative of Class 3. (**E**) Diagram of DTC–Sh1 relationship in distal end of a typical wild-type gonad (DTC magenta, Sh1 green) for phenotypic Classes 1–3. Class 1: greater than 8 µm (two cell diameters) separates the proximal most extent of the DTC (vertical line topped with magenta dot) and the distal most border of Sh1 (vertical line topped with green dot) as measured in slice-by-slice analysis of Z series; Class 2: ≤8 µm separates DTC and Sh1 extensions or interdigitation of DTC and Sh1 is seen up to the position of the average DTC process length (vertical line topped with blue dot); Class 3: distal position of Sh1 is distal to the average DTC process length. A diagram of all measurements taken for live fluorescent images is shown in *Figure 1—figure supplement 1*. (**F**) The proportion of gonads examined that fall into the phenotypic classes indicated. (**G**) Plot showing the distance between the average DTC process length (blue dots), the longest DTC process (magenta dots) and the distal-most border of Sh1 (green dots); each trio of dots joined by a dark blue line represents the data for a single gonad, analyzed as a confocal maximum projection. Scale bars are 10 µm. Strains used are DG5020, DG5026, and DG5029; n for each is indicated in panel **F**. See *Supplementary file 1* for complete genotypes.

The online version of this article includes the following source data and figure supplement(s) for figure 1:

**Source data 1.** Source data for *Figure 1C–D* and *Figure 1F–G*.

**Figure supplement 1.** Consistent trends in Sh1 and DTC positions are observed with multiple markers.

**Figure supplement 1—source data 1.** Source data for *Figure 1—figure supplement 1B–E*.

**Figure supplement 2.** Snapshot of transmission electron micrograph (TEM) reconstruction and guide to *Videos 4 and 5*.

of stem cells, such that a daughter cell in contact with Sh1 enters the differentiation pathway while the other, in contact with the DTC, remains a stem cell (*Gordon et al., 2020*). If oriented cell fate-determining germ cell divisions occur in this way, the expectation would be that germ cells undergoing such a division should be in contact with both the DTC and Sh1, and that closely spaced daughter germ cells that have recently undergone such a division would remain in contact with one or the other. The model predicts that there should be few, if any, germ cells that contact neither the DTC nor Sh1. If Sh1 drives cell fate-determining germ cell divisions, another expectation would be that altering the position of Sh1 in the distal–proximal direction should similarly alter the position of the germline stem cell fate decision. However, given that much of the analysis was performed using an INX-8 fusion protein marker that could conceivably alter the position of Sh1, and given that the precise relationship between the position of Sh1 vis-à-vis the border of the stem cell pool was not investigated, we wished to determine how hypomorphic innexin alleles alter the position of Sh1 and whether the position of Sh1 influences the border of the germline stem or progenitor pools.

In short, we characterize the natural variability of Sh1 position and confirm the presence of bare regions in gonads of early adult wild-type hermaphrodites. We show that reducing soma–germline gap-junction coupling causes Sh1 to be mispositioned distally. In addition, we show that the fusion protein used in the previous study (*Gordon et al., 2020*) encodes a dominant antimorphic allele of *inx-8* that also causes distal mispositioning of Sh1. Finally, we determine that when bare regions are absent due to Sh1 distal mispositioning, the borders of the stem or progenitor cell pools are not markedly altered. Together, these results demonstrate that while innexin function can dramatically impact the position of the distal border of Sh1, this position of Sh1 is not relevant for GLP-1/Notch signaling nor for germ cells to exit the stem cell pool.

## Results

### There is natural variability in the position of the distal border of Sh1, but gonads of wild-type hermaphrodites display bare regions where germ cells contact neither the DTC nor Sh1

As detailed below, we found that Sh1 is somewhat variably positioned in wild-type young adult hermaphrodites. We confirmed the existence of bare regions between Sh1 and the DTC, characterized by germ cells that do not contact Sh1 or the DTC.

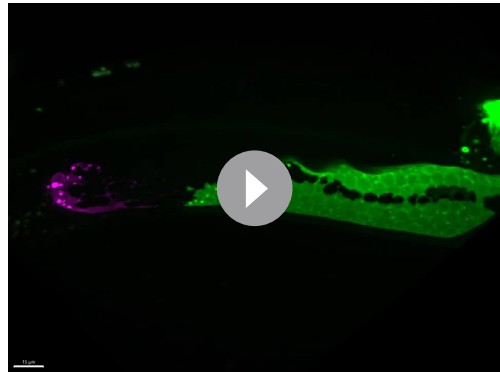

**Video 1.** Confocal stack and 3D rendering of representative Class 1 gonad. Strain DG5020.
https://elifesciences.org/articles/74955/figures#video1

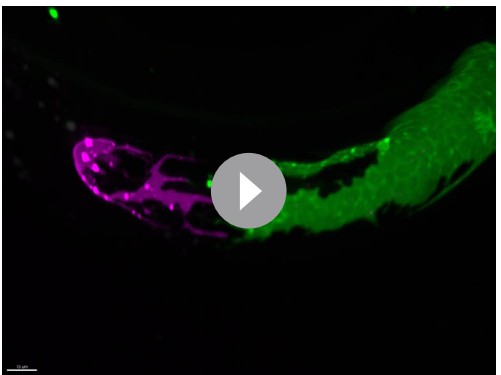

**Video 2.** Confocal stack and 3D rendering of representative Class 2 gonad. Strain DG5020. https://elifesciences.org/articles/74955/figures#video2

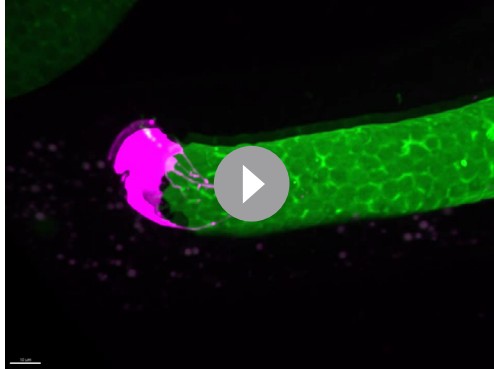

**Video 3.** Confocal stack and 3D rendering of representative Class 3 gonad. Strain DG5026. https://elifesciences.org/articles/74955/figures#video3

To better characterize the natural variability in the position of the distal border of Sh1 in the wild type, and to facilitate comparisons among different genotypes and markers, we binned gonads into one of three phenotypic classes based on the position of Sh1 (*Figure 1C–F*; *Videos 1–3*; *Figure 1— figure supplement 1*). To avoid ambiguity that results from the analysis of maximum projection images, we analyzed all gonads in 3 dimensions (based on 0.5 µm optical Z-series slices). We designated as Class 1, those gonads in which > 8 µm (a distance covered by 2 germ cell diameters) separated the DTC and Sh1. Class 2 includes those gonads in which Sh1 and DTC processes come within 8 µm or in which Sh1 extensions intercalate with DTC processes, up to the position of the mean length of DTC processes. Class 3 gonads are those in which the border of Sh1 is markedly shifted distally such that it lies between the distal end of the gonad and the mean DTC process length.

Using *bcIs39[lim-7p::ced-1::GFP]* to visualize Sh1, a transgene that produces a CED-1::GFP fusion protein that fully rescues the sheath cell function of CED-1 to engulf germ cell corpses (*Zhou et al., 2001*), we found that a majority of gonads (70%) are in Class 1, confirming the existence of bare regions. Indeed, in many cases, the distance between the proximal-most DTC measurement and the distal-most Sh1 measurement exceeds 25 µm (*Figure 1G*; *Figure 1—figure supplement 1D*). The remaining gonads are in Class 2.

Using three additional Sh1 markers (and one alternate DTC marker), we found that depending

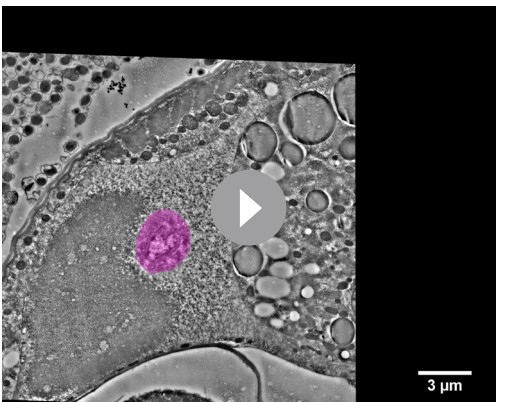

**Video 4.** Serial thin section movie of every third section from the DTC to the distal extensions of Sh1 in a young adult hermaphrodite posterior gonad arm. Germ cell nuclei are marked in yellow if the corresponding germ cells receive direct contact from a DTC or Sh1 process, or in salmon if they do not. The soma and processes of the DTC are filled in pink. Filopodial extensions of the somatic sheath are filled in green. Individual images are composed from multi-panel montages (mostly 3x4 montages). Scale bar is 3 µm. https://elifesciences.org/articles/74955/figures#video4

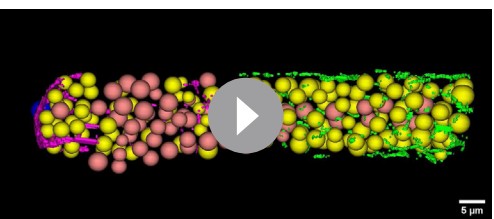

**Video 5.** Model of the distal gonad. The DTC, its processes and fragments are in pink. Nearest sheath cell bodies lie out of frame proximally, but their filopodial extensions are shown in green, extending towards the DTC. Germ cell nuclei are modelled as spheres; the nuclei of those germ cells that do not come in contact with either DTC or Sh1 are colored in salmon, while germ cells with direct contact are shown in yellow. The true volume of each germ cell is larger than is shown here because each sphere is centered only on the cell nucleus. Scale bar is 5 µm. https://elifesciences.org/articles/74955/figures#video5

on the markers used, 50–90% of gonads fall into Class 1, while Class 2 make up the remainder (*Figure 1—figure supplement 1*). We note that both Class 1 and Class 2 gonads contain bare regions with spacing of germ cell nuclei (as seen by differential interference contrast microscopy) such that their surrounding cell boundaries are not plausibly in contact with the DTC or Sh1. We did not observe Class 3 gonads in any wild-type worms examined.

We also analyzed serial distal-to-proximal transmission electron micrographs (TEM) of a gonad from a wild-type, unmarked young adult worm fixed using high-pressure freezing and freeze-substitution freezing methods (*Figure 1—figure supplement 2*; *Videos 4 and 5*; Materials and methods). Focusing on an 80 μm region from the distal end to the distal extensions of Sh1, we followed DTC processes (and their fragments) and Sh1 filopodial extensions among the tightly packed germ cells (see Materials and methods). We observed germ cells that contact neither a DTC process (nor fragment thereof) nor filopodial extensions of Sh1, and thus represent bare regions. Therefore, this TEM analysis is consistent with the confocal analysis above using cell-type-specific markers in live worms, and it corroborates prior ultrastructural observations made on chemically fixed TEM preparations (*Hall et al., 1999*).

## Distal somatic gonad architecture is dictated by both somatic- and germline-expressed innexins

Previously, in fixed preparations, we observed that the distal edge of Sh1 was shifted nearly to the distal end of the gonad in worms bearing a hypomorphic mutation in the germline innexin *inx-14(ag17)* (*Starich et al., 2014*). We further investigated the position of Sh1 in *inx-14(ag17)* using live imaging of intact young adult hermaphrodites bearing contrasting markers for Sh1 and the DTC (see Materials and methods for details on markers used).

In contrast to *inx-14(+)*, the distal border of Sh1 in *inx-14(ag17)* extends to the distal end of the gonad, well distal to the average position of the DTC processes (*Figure 1C–E*). The vast majority of *inx-14(ag17)* gonads fall into Class 3 (*Figure 1*), with consistent results from three different markers of Sh1 (*Figure 1—figure supplement 1B–E*). Although our classification scheme references Sh1 relative to DTC position, we do not attribute this gross positional alteration of Sh1 to a change in the DTC. Instead, within strains with similar markers, the change in the *inx-14* genotype does not significantly change average DTC mean process length or the length of the longest DTC process, but it does have a significant effect on Sh1 distal border position (*Figure 1G*; *Figure 1—figure supplement 1E*).

To determine whether the position of Sh1 is shifted distally upon reduction of the somatic innexins, we investigated the position of Sh1 relative to the distal end of the gonad in a well-characterized compound reduction-of-function (rf) *inx-8* mutant (*Starich et al., 2020*; *Figure 1B–G*). We found that Sh1 in worms bearing one partially functional somatic gonad innexin encoded by *inx-8(tn1513tn1555)* in an otherwise null *inx-9(ok1502)* background (hereafter referred to as '*inx-8(rf)*'; *Starich et al., 2020*) is also distally positioned, similar to *inx-14(ag17)* (*Figure 1C–E*). Thus, Sh1 is positioned distally as a result of reducing the function of either germ line or somatic innexins in the distal gonad.

## An mKate2::INX-8 fusion causes a distal shift in the border of Sh1

We extended our analysis to *inx-8(qy78[mKate2::inx-8])*, an allele that encodes an N-terminal fusion protein of mKate2 and INX-8 that was used to mark Sh1 in a prior study (*Gordon et al., 2020*; *Figure 2A*). As detailed below, we found that—like *inx-14(ag17)* and *inx-8(rf)*—*inx-8(qy78[mKate2::inx-8])* caused a distal shift in the position of Sh1 (*Figure 2B–F*). We documented this shift both by anti-INX-8 antibody (*Starich et al., 2014*) detection in otherwise unmarked strains (*Figure 2B–C*), as well as in live imaging of marked strains (*Figure 2D–F*).

We used anti-INX-8 antibody staining to compare the distal border of Sh1 in the unmarked wild type and in *inx-8(qy78[mKate2::inx-8])* gonads (N2 and DG5063, respectively; see *Supplementary file 1*; *Figure 2—figure supplement 1*). The mean distal boundary of INX-8 staining in the wild type lies at ~80 μm from the distal end of the gonad, while the mean distal boundary of INX-8 staining in worms carrying *inx-8(qy78[mKate2::inx-8])* lies at ~50 μm from the distal end, and the difference between these means is significant (*Figure 2B–C*). Moreover, in 89% of wild-type gonads (n=28), the distal border of Sh1 is greater than 60 μm (~15–16 cell diameters) from the distal end of the gonad, while in 79% of *inx-8(qy78[mKate2::inx-8])* gonads (n=32) the distal border of Sh1 is less than 60 μm from the distal end (*Figure 2B–C*).

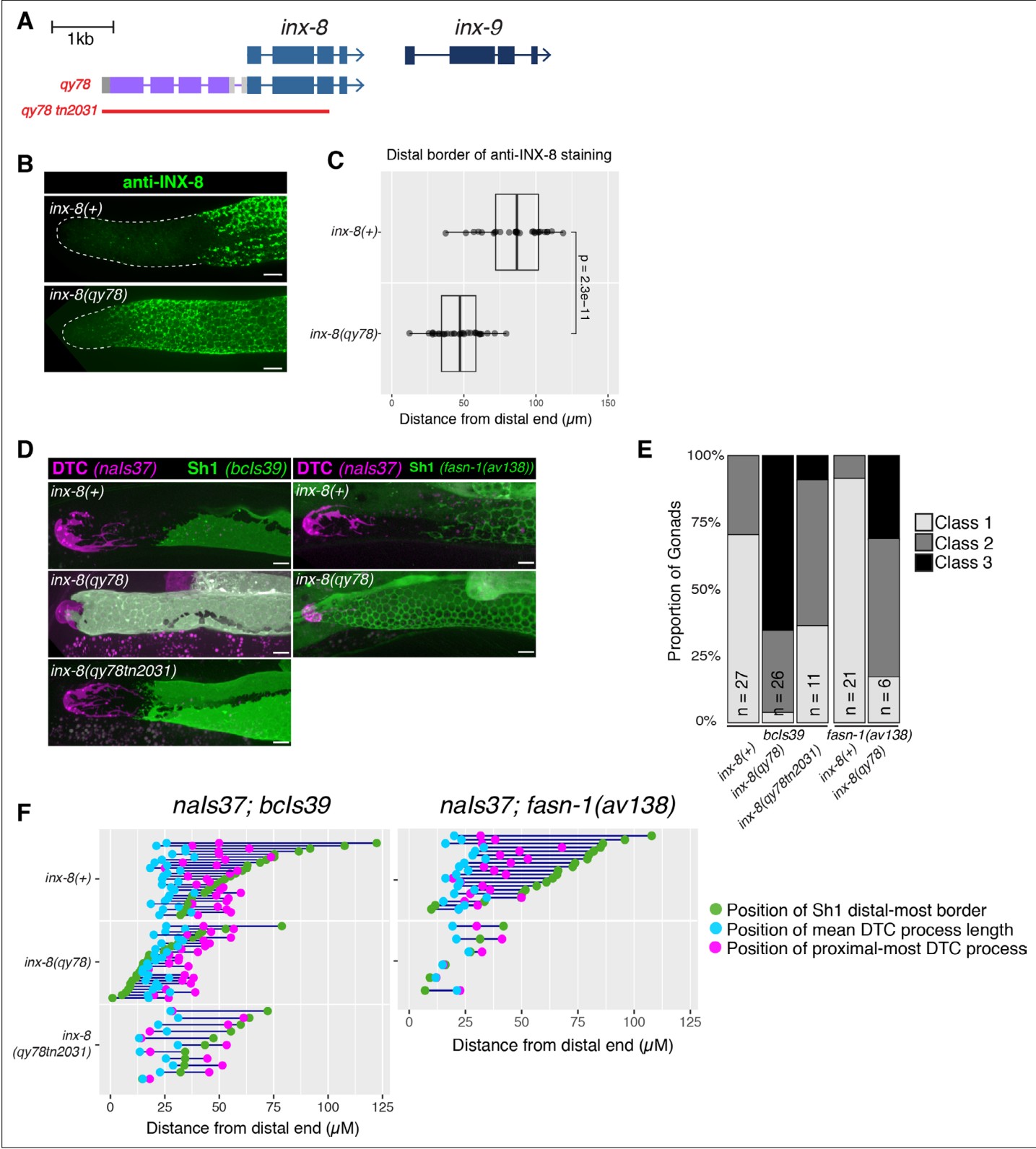

**Figure 2.** N-terminal fusion of mKate2 to INX-8 generates an INX-8 protein that alters somatic gonad morphology. (**A**) Schematic diagram showing the genetic manipulations used in this section. *inx-8(qy78[mKate2::inx-8])* was created by placing mKate2 in-frame with the N-terminus of INX-8 (*Gordon et al., 2020*). *inx-8(qy78tn2031)* was created by deleting the *inx-8* coding region and mKate2 moiety in the *inx-8(qy78)* background. (**B**) Representative distal gonads stained with anti-INX-8 antibody; Top: N2 wild type, Bottom: DG5063 *inx-8(qy78[mKate2::inx-8])*. See legend to *Figure 2—figure*

*Figure 2 continued on next page*

*Figure 2 continued*

*supplement 1* for further details. (**C**) Dot plot with overlaid quantile box plots showing the distance from the distal end of the gonad to the Sh1 distal border for each genotype. Each dot represents a single gonad of that genotype. p value was calculated using Student's *t*-test. (**D**) Fluorescent micrographs of live animals with the DTC marked by *naIs37[lag-2p::mCherry-PH]* and Sh1 marked by *bcIs39[lim-7p::ced-1::gfp]*. Top: wild-type with markers only. Middle: *inx-8(qy78[mKate2::inx-8])*. Bottom: *inx-8(qy78tn2031)*. (**E**) The proportion of gonads examined that fall into the phenotypic classes indicated. (**F**) Quantitative dot-plots as in *Figure 1*; data for wild-type (marker only) strain in **E** and **F** is the same as in *Figure 1*. Each pair of dots connected by a line represents data from a single gonad. Scale bar 10 µm. Strains used were N2, DG5063, DG5020, DG5131, DG5229, DG5320, DG5378; n for each is indicated in panel **E**. See *Supplementary file 1* for complete genotypes.

The online version of this article includes the following source data and figure supplement(s) for figure 2:

**Source data 1.** Source data for *Figure 2B and C*.

**Source data 2.** Source data for *Figure 2D–F*.

**Figure supplement 1.** INX-8 antibody staining overlaps with mKate2 fluorescence in *inx-8(qy78[mKate2::inx-8])* gonads.

**Figure supplement 2.** The *inx-8(qy78[mKate2::inx-8])* allele behaves as a dosage-sensitive antimorph.

**Figure supplement 2—source data 1.** Source data for *Figure 2—figure supplement 2B–E*.

**Figure supplement 3.** Direct comparison of *inx-8(qy78[mKate2::inx-8])* and *lim-7p*-driven transgenes as markers for Sh1.

**Figure supplement 3—source data 1.** Source data for *Figure 2—figure supplement 3*.

Similar results were obtained using live imaging. Using *bcIs39* to visualize Sh1, no Class 3 gonads (n=27) are observed in the wild type, compared to 65% (n=26) in *inx-8(qy78[mKate2::inx-8])* (*Figure 2D–E*). The same mKate2::INX-8 fusion, independently generated in an *inx-9(ok1502)* null mutant background (i.e. *inx-8(qy102[mKate2::inx-8]*, see Materials and methods *Gordon et al., 2020*), was also observed to shift Sh1 distally in *inx-8(qy102[mKate2::inx-8]) inx-9(ok1502)* double mutants. In contrast, loss of *inx-9* alone does not significantly affect Sh1 position (*Figure 2—figure supplement 2*). If mKate2::INX-8 were innocuous, the *inx-8(qy102[mKate2::inx-8])* allele in the absence of *inx-9* (*inx-9(1502)*) should have had no effect on Sh1 position.

To explain the observed distal Sh1 border shift of mKate2::INX-8 compared to previous descriptions of distal gonad architecture, *Gordon et al., 2020* postulated that widely used markers of Sh1 driven by the *lim-7* promoter may not mark the entirety of the cell, excluding the distal-most areas from detection. Although anti-INX-8 antibody staining clearly indicates a more proximal position of Sh1 in otherwise unmarked wild-type strains and an evident distal Sh1 shift in mKate2::INX-8, we wished to further determine whether the distal shift of Sh1 we observed in *inx-8(qy78[mKate2::inx-8])* and *inx-14(ag17)* backgrounds reflected a disparity between the expression patterns of mKate2::INX-8 and *lim-7p*-driven markers. First, we consistently observe a dramatic distal shift of Sh1 in *inx-14(ag17)* using *lim-7* promoter-driven transgenes (*bcIs39* and *tnIs5*) as well as using a functional endogenously tagged allele of the fatty acid synthase gene *fasn-1*, [*fasn-1(av138[fasn-1::GFP])*] (*Figure 1D*; *Figure 1—figure supplement 1*). We next examined the overlap between the mKate2::INX-8 and *lim-7p*-driven markers in strains expressing both *inx-8(qy78[mKate2::inx-8])* and GFP markers encoded by *tnIs6* [*lim-7p::gfp*] or *bcIs39* [*lim-7p::ced-1::gfp*]. In over 85% of gonad arms examined (n=20 and 26, respectively), the overlap between mKate2::INX-8 and GFP was complete (*Figure 2—figure supplement 3*). In both *tnIs6* [*lim-7p::gfp*] or *bcIs39* [*lim-7p::ced-1::gfp*], the remaining 15% of gonads displayed either reduced Sh1 expression or expression in one, but not both, cells of the Sh1 pair, which may result from stochastic transgene downregulation. Finally, we binned these gonads into phenotypic Classes 1–3 based on measurements taken while observing either mKate2 *or* GFP expression (*Figure 2—figure supplement 3D*). The number of Class 3 gonads is somewhat underestimated with the GFP markers. Nevertheless, we observe a striking distal shift of Sh1 in the *inx-14* and *inx-8* mutants bearing these markers. In sum, if the GFP markers were not marking the distal-most part of Sh1, we should not have been able to so readily detect the abnormal distal displacement of Sh1 in *inx-14(ag17)*, *inx-8(rf)* or *inx-8(qy78[mKate2::inx-8])* strains bearing these markers.

We conclude that despite variability in the exact position of the Sh1 border, and despite minor differences in transgene expression, bare regions between the Sh1 and the DTC are present in the wild type, and they are greatly diminished due to a distal shift in the position of Sh1 in worms bearing reduced *inx-14* or *inx-8* function or bearing the mKate2::INX-8 fusion.

## An mKate2::INX-8 fusion behaves as a dominant dosage-sensitive antimorph

Based on our observations that *inx-8(qy78[mKate2::inx-8)]* gonads display a distally shifted border of Sh1, we hypothesized that this allele might encode an INX-8 protein that poisons innexin complexes thereby interfering with the channel and/or adhesion functions important for Sh1 positioning. If this were due to a toxic effect of the mKate2::INX-8 fusion protein on gap junctions, we would predict that the distal shift of Sh1 would be dependent on the presence of the INX-8 coding region. To test this hypothesis, we used CRISPR-Cas9 genome editing to generate *inx-8* null alleles in both the *inx-8(qy78[mKate2::inx-8])* and wild-type genetic backgrounds. We generated deletions with identical breakpoints in the *inx-8* locus in both genetic backgrounds [i.e. *inx-8(qy78tn2031)* and *inx-8(tn2034)*] starting 136 bp upstream of the wild-type *inx-8* ATG start codon and extending 221 bp into *inx-8* exon 3 (*Figure 2* and *Figure 2—figure supplement 2*). In the *inx-8(qy78[mKate2::inx-8])* context, this deletion also removes the mKate2 moiety. These deletions are expected to constitute *inx-8* null alleles because, in addition to removing the start codon, they delete amino acids 1–349 (out of 382 amino acids), including virtually all residues essential for spanning the plasma membrane and forming a channel (*Starich and Greenstein, 2020*). Consistent with our hypothesis, the deletion mutant *inx-8(qy78tn2031)* displays a significantly lower frequency of Class 3 gonads than the original *inx-8(qy78[mKate2::inx-8])* allele. Sh1 in the *inx-8(qy78tn2031)* deletion mutant is positioned

**Table 1.** Brood sizes and embryonic lethality measurements for selected strains.

| Strain | Relevant genotype* | Brood (n)† | %Lethality (n) ‡ | Sh1 mispositioned§ |
|---|---|---|---|---|
| N2 | Wild type | 298.6±42.1 (24) | 0.4±0.4 (7194) | no |
| DG5063 | *inx-8(qy78)* ¶ | 221.6±23.6 (25) | 9.0±4.6 (6087) | yes |
| NK2571 | *inx-8(qy78); cpIs122* | 212.8±27.5 (23) | 6.2±3.0 (5217) | (yes**) |
| DG5059 | *inx-9(ok1502)* | 322.7±33.0 (25) | 0.2±0.3 (8081) | (no**) |
| DG5064 | *inx-8(qy102) inx-9(ok1502)*†† | 211.1±47.2 (25) | 1.1±1.1 (5327) | (yes**) |
| NK2576 | *inx-8(qy102) inx-9(ok1502); cpIs122* | 209.5±52.7 (23) | 2.3±1.2 (4928) | (yes**) |
| DG5250 | *inx-8(qy78tn2031)* ‡‡ | 260.0±25.9 (25) | 0.1±0.2 (6504) | (no**) |
| DG5251 | *inx-8(tn2034)* ‡‡ | 301.6±37.8 (25) | 0.4±0.4 (7570) | (no**) |
| DG5270 | *inx-14(ag17)* | 253.2±28.1 (25) | 0.4±0.6 (6356) | (yes**) |
| DG5070 | *inx-14(ag17); inx-8(qy78)* | 106.6±19.4 (25) | 23.7±8.0 (3510) | n.d. |
| DG5380 | *bcIs39* | 282.6±34.2 (25) | 0.4±0.5 (7091) | n.d. |
| DG5020 | *naIs37; bcIs39* | 237.5±46.5 (24) | 1.0±1.4 (5753) | no |
| DG5320 | *fasn-1(av138::gfp) naIs37* | 342.4±30.3 (25) | 0.4±0.4 (8591) | no |
| DG5367 | *inx-14(ag17) fasn-1(av138::gfp) naIs37* | 256.0±26.6 (23) | 0.2±0.2 (5897) | yes |
| DG5378 | *fasn-1(av138::gfp) naIs37; inx-8(qy78)* | 245.4±43.7 (23) | 5.3±3.2 (5947) | yes |

*Full genotypes in **Supplementary file 1**; genotypes *inx-8(qy78[mKate2::inx-8])* and *inx-8(qy102[mKate2::inx-8])* are abbreviated as "*inx-8(qy78)*" and "*inx-8(qy102)*" and *fasn-1(av138[fasn-1::GFP])* as *fasn-1(av138::gfp)*.

†Brood size ± SD, n = number of broods counted.

‡Percent lethality ± SD, n = number of embryos scored.

§yes/no indication of distal shift in the position of Sh1; relevant measurements in **Figure 1**, **Figure 2**, and their related supplements.

¶Derived from NK2571 *inx-8(qy78); cpIs122 [lag-2p::mNeonGreen::plcdeltaPH]* (**Gordon et al., 2020**).

**These particular strains were not scored for Sh1 position. However, where equivalent *inx* genotypes were analyzed, the Sh1 phenotype is given in parenthesis; see **Figure 1**, **Figure 2**, and their related supplements.

††Derived from NK2576 *inx-8(qy102) inx-9(ok1502); cpIs122* (**Gordon et al., 2020**).

‡‡Equivalent *inx-8(null)* CRISPR-Cas9 deletions generated in *inx-8(qy78[mKate2::inx-8])* or in wild-type genetic backgrounds.

The online version of this article includes the following source data for table 1:

**Source data 1.** Brood size and embryonic lethality data for **Table 1**.

more proximally, and therefore, like the wild type, exhibits bare regions that are lacking in *inx-8(qy78[mKate2::inx-8])* mutants (*Figure 2D–F*). In addition, *inx-8(qy78tn2031)* exhibits a nearly normal brood size and suppresses the low-level embryonic lethality observed in the *inx-8(qy78)* starting strain (*Table 1*). Likewise, the identical deletion generated in the wild-type genetic background, *inx-8(tn2034)*, also exhibits the wild-type Sh1 position (*Figure 2—figure supplement 2*), with a normal brood size and negligible embryonic lethality (*Table 1*). However, the observation of a small percentage of Class 3 gonads in *inx-8(qy78tn2031)* and *inx-8(tn2034)* animals but not in the wild type (*Figure 2—figure supplement 2*), suggests that the position of Sh1 is sensitive to the dosage of somatic gap junction hemichannels. In fact, the *inx-8(qy78tn2031)/+* heterozygote reveals apparent haploinsufficiency for this phenotype (*Figure 2—figure supplement 2*).

We made several additional observations from a brood-size and embryonic-viability analysis (*Table 1*). First, the marker transgenes do not have an appreciable effect on brood size or embryonic lethality in our hands. Second, genetic interactions exist between *inx-8(qy78[mKate2::inx-8])*, *inx-14(ag17)* and the *inx-9(ok1502)* null mutation. Specifically, *inx-14(ag17)* enhances both the reduced brood size and embryonic lethality of *inx-8(qy78[mKate2::inx-8])*; whereas *inx-9(ok1502)* weakly suppresses the embryonic lethality defect (*Table 1*). Third, in the course of this analysis, we noted that strains bearing *inx-8(qy78[mKate2::inx-8])* displayed highly variable population growth dynamics that we attribute to stress. Although the relevant stressor has yet to be determined, we observed this effect over multiple generations after starvation or transport. Given our previous finding that key metabolites such as malonyl-CoA are transported through these junctions to support embryonic growth, this finding deserves future scrutiny.

Finally, to assess more fully the nature of the *inx-8(qy78[mKate2::inx-8])* allele with respect to Sh1 position, we generated and imaged worms bearing this allele in heterozygous and hemizygous configurations (*Figure 2—figure supplement 2*; we abbreviate *inx-8(qy78[mKate2::inx-8])* to '*qy78*' and null to (0), below, to aid comparison between genotypes). We infer loss-of-function behavior since other loss-of-function mutations affecting soma-germline gap junctions, such as *inx-14(ag17)*, also cause a distal shift in the Sh1 border. Nevertheless, *inx-8(qy78)* is dominant: a substantial proportion of gonads in *qy78/+* are in Class 3, similar to *qy78/qy78*, but distinct from deletion heterozygotes [*inx-8(qy78tn2031/+)*]. However, in a hemizygous configuration (*qy78/0*), the Sh1 distal displacement phenotype is largely suppressed. Thus, *qy78* acts as a dosage-sensitive antimorph. We speculate that the fusion protein produced by *inx-8(qy78[mKate2::inx-8])* can form dysfunctional junctions on its own or with INX-8(+), but not when it is present below a certain abundance threshold. As also mentioned above, the *inx-8* deletion allele *qy78tn2031*, while largely suppressing the Sh1 distal mispositioning defect, displays a weak dominant haploinsufficient behavior consistent with dosage sensitivity. Alternatively, this behavior could result from genetic positional effects that impinge on nearby *inx-9* function and for which Sh1 position is particularly sensitive.

## The distal position of Sh1 does not influence the position of the stem cell pool border

A recent model proposed that the position of the Sh1 border influences the stem/non-stem decision in underlying germ cells (*Gordon et al., 2020*). However, because the previous study did not examine the position of stem or progenitor cells, and because the model was based on results using the antimorphic *inx-8(qy78[mKate2::inx-8])* allele, we investigated this relationship.

In its simplest form, the model proposed by *Gordon et al., 2020* predicts that when the distal edge of Sh1 is positioned distally, the stem/non-stem border should similarly shift distally. The SYGL-1 protein serves as a stem cell marker as *sygl-1* is a direct transcriptional target of GLP-1/Notch in the germ line (*Chen et al., 2020*; *Kershner et al., 2014*; *Lee et al., 2019*; *Lee et al., 2016*; *Shin et al., 2017*). We analyzed the proximal extent of the pool of SYGL-1-positive cells bearing a well-characterized OLLAS epitope tag on SYGL-1 (*Shin et al., 2017*) and compared that boundary relative to the distal Sh1 border (*Figure 3* and Materials and methods). In the case of *inx-14(ag17)*, although the distal border of Sh1 was shifted drastically and significantly, there was no significant change in the position of the border of the SYGL-1-positive stem cell pool. In the case of the *inx-8(qy78[mKate2::inx-8])* allele, the border of the SYGL-1-positive pool was marginally shifted distally relative to the wild type, though not commensurate with the extent to which Sh1 shifted distally in this background (*Figure 3C*). Furthermore, the marginal shift of the border of the stem cell pool in *inx-8(qy78[mKate2::inx-8])* was

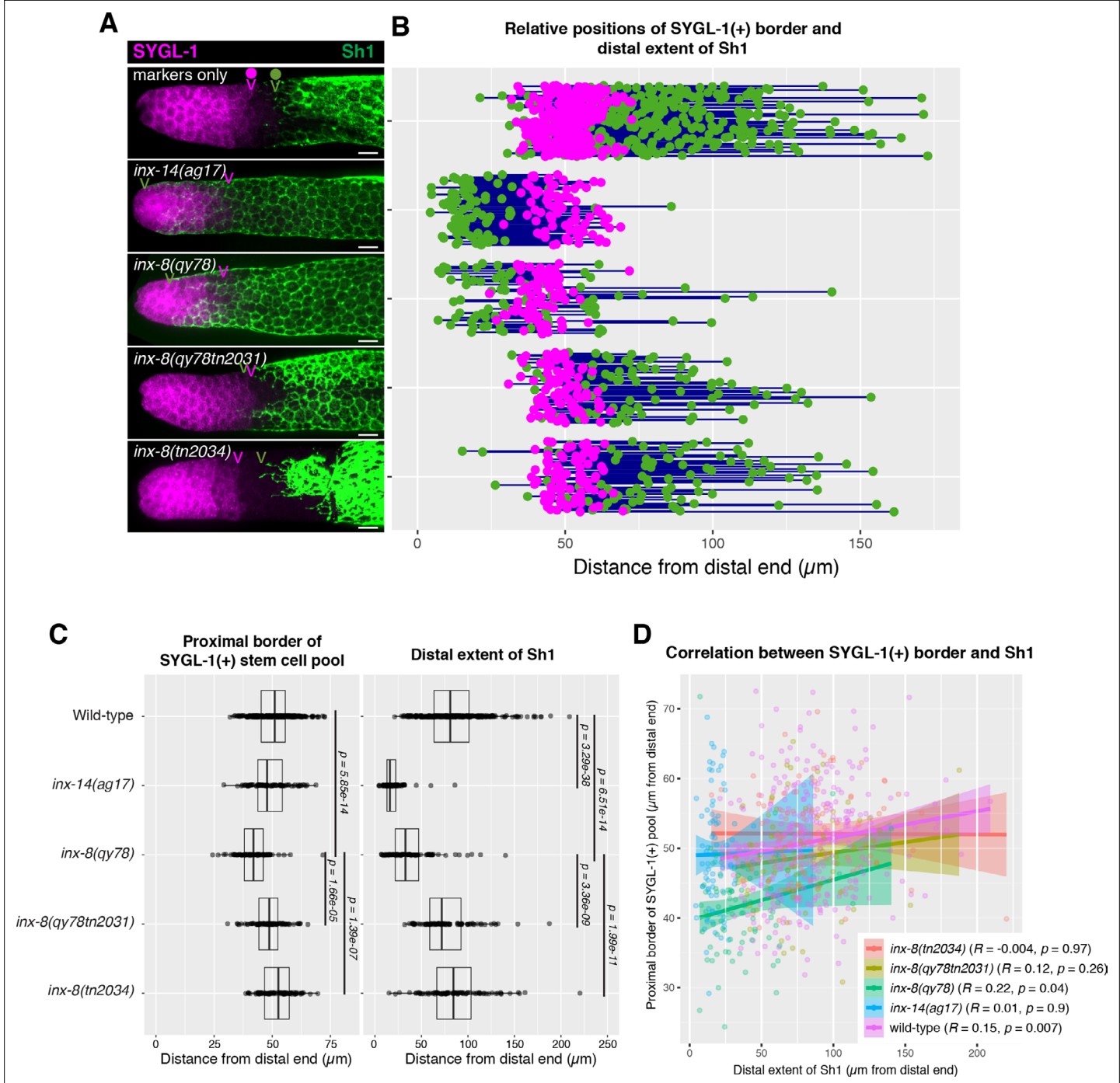

**Figure 3.** The position of the proximal border of the SYGL-1-positive stem cell pool does not correlate with the position of Sh1. (**A**) Fluorescence confocal maximum projection of surface images (~7 μm depth) of fixed, dissected gonads showing the SYGL-1-positive stem cell pool marked in magenta and the sheath cell marked in green. Magenta caret represents the location of the proximal border of the SYGL-1-positive stem cell pool. Green caret represents the distal edge of Sh1. (**B**) Quantitative graph with magenta dots representing the proximal border of SYGL-1::OLLAS expression and green dots representing the distal reach of Sh1. Each pair of dots connected by a line represents the data for a single gonad. (**C**) Dot plot with overlaid box plots showing the distance and quantiles of the SYGL-1-positive stem cell pool and distal extent of Sh1 for each genotype. Each dot represents a single specimen of that genotype. p values were calculated using Student's *t*-test. (**D**) Scatterplot showing lack of correlation between the proximal extent of SYGL-1 expression and the distal reach of Sh1. Scale bars are 10 μm. Strains used, top to bottom as indicated on figure panels **A** and **C**, and (n) number of gonads are DG5136 (320), DG5150 (83), DG5181 (88), DG5248 (86), and DG5249 (88). See ***Supplementary file 1*** for complete genotypes.

The online version of this article includes the following source data and figure supplement(s) for figure 3:

*Figure 3 continued on next page*

*Figure 3 continued*

**Source data 1.** Source data for *Figure 3* and *Figure 3—figure supplement 1*.

**Figure supplement 1.** The position of the proximal border of the SYGL-1-positive stem cell pool does not correlate with the position of Sh1 when data are shown in cell diameters.

**Figure supplement 2.** Proximal end of the SYGL-1 zone assessed visually and from protein levels.

suppressed when *inx-8* was deleted in both *inx-8(qy78tn2031)* and *inx-8(tn2034)*. The marginal shift in the border of the stem cell pool is not observed in *inx-14(ag17)*, suggesting that such a defect in *inx-8(qy78[mKate2::inx-8])* is due to the altered function of mKate2::INX-8, rather than the position of Sh1 per se (*Figure 3A–C*). To detect any subtle correlation between the proximal border of the SYGL-1 pool and the distal border of Sh1, we plotted these against one another and computed an R value (*Figure 3D*). By Pearson correlation, there is no significant relationship in any genotype examined between the position of the Sh1 border and the position of the SYGL-1-positive stem cell pool border. There is no significant relationship regardless of whether measurements are made in microns (*Figure 3*) or cell diameters (CD; *Figure 3—figure supplement 1*).

The recent model also proposed that Sh1 controls spindle orientation at the stem/progenitor border. However, we found that in the wild-type (marker-only) strain, the distal border of Sh1 was proximal to the proximal SYGL-1-positive border in 86% of the gonads (n=320), with a distance between them of ≥5 CD in 67% of the gonads (*Figure 3* and *Figure 3—figure supplement 1*). As mentioned above, this 5 CD distance is inconsistent with the hypothesis that Sh1 is controlling spindle orientation at the stem cell border as such control would be expected to occur over a distance of only 1–2 cell diameters.

We conclude that (1) there is no correlation between the position of the distal border of Sh1 and the proximal border of the SYGL-1-positive stem cell pool, and (2) if spindle-oriented divisions occur at the Sh1 border, they are not influencing stem cell fate.

## The distal position of Sh1 does not influence the position of the progenitor pool border

Although we found no correlation between the stem cell pool border and Sh1 position, we wondered whether altered Sh1 position might nevertheless influence the position of the border between the progenitor zone (PZ) and the transition zone that marks overt meiotic entry. In the wild type, consistent with our previous analysis in live worms, we found that the distal position of Sh1 is somewhat variable and can be either distal or proximal of the PZ border, using the length of the CYE-1-positive region to define the PZ border, following CYE-1 and pSUN-1(S8) co-staining (*Figure 4* and Materials and methods; *Mohammad et al., 2018*). We found that although there is a subtle shift in the PZ border in *inx-8(qy78[mKate2::inx-8])* and *inx-14(ag17)*, it does not correlate with the dramatic shift in Sh1 position seen in these mutants (*Figure 4*). Again, we saw no significant correlation between Sh1 position and the position of the PZ border (*Figure 4D*). As with the SYGL-1 analysis, the relationships are similar when measurements are made in microns (*Figure 4*) or in CD (*Figure 4—figure supplement 1*), noting that differences between genotypes are more apparent in the micron measurements since cells are slightly larger more proximally in the PZ.

In sum, the border between the PZ and the transition zone is not markedly altered relative to the dramatic distal displacement of the Sh1 border in *inx-14(ag17)* or in *inx-8(qy78[mKate2::inx-8])*, suggesting that the position of Sh1 has no bearing on the position of meiotic entry.

## Discussion
### Contact with Sh1 does not dictate the stem cell pool border

Our studies show that impaired innexin function distally mispositions Sh1, but that this displacement does not similarly shift the proximal border of the adult stem cell pool (*Figure 5*). We show that bare regions of variable size normally exist between the DTC and Sh1, in which germ cells lack contact with the DTC or Sh1, but that these bare regions are diminished or lost with reduced innexin activity either in the soma or the germ line. While our studies rule out a role for Sh1 in stem cell fate, they demonstrate a role for innexins in somatic gonad architecture.

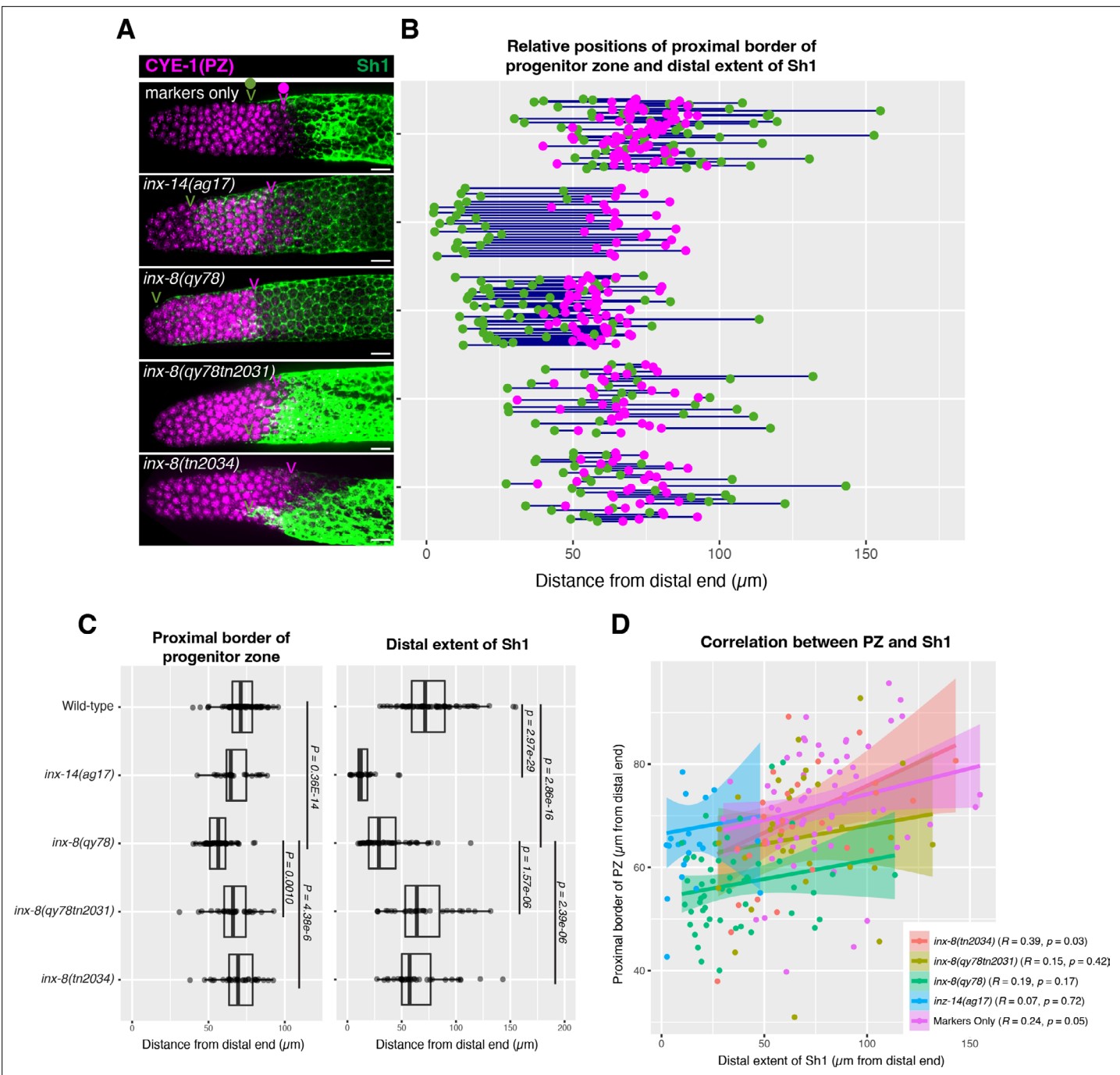

**Figure 4.** The position of the proximal border of the progenitor zone does not correlate with the position of Sh1. (**A**) Fluorescence confocal maximum projection of surface images (~7 μm depth) of fixed, dissected gonads showing the progenitor pool marked in magenta and the sheath cell marked in green. All measurements were made by examining optical sections through the entire depth of the gonad. Magenta caret represents the location of the proximal border of the CYE-1-positive, pSUN-1-negative progenitor pool (pSUN-1 staining not shown; see Materials and methods). Green caret represents the distal edge of Sh1. (**B**) Quantitative graph with magenta dots representing the proximal extent of the CYE-1 staining and green dots representing the distal reach of Sh1. Each pair of magenta and green dots connected by a line represents the data for a single gonad. (**C**) Dot plot with overlaid box plots showing the distance and quantiles of the progenitor pool and distal extent of Sh1 for each genotype. Each dot represents a single specimen of that genotype. p values were calculated using Student's *t*-test. (**D**) Scatterplot showing lack of correlation between the proximal PZ border and Sh1 position. Scale bars are 10 μm. Strains used, top to bottom as indicated on figure panels **A** and **C**, and (n) number of gonads are DG5136 (72), DG5150 (26), DG5181 (52), DG5248 (30), and DG5249 (32). See *Supplementary file 1* for complete genotypes.

The online version of this article includes the following source data and figure supplement(s) for figure 4:

*Figure 4 continued on next page*

*Figure 4 continued*

**Source data 1.** Source data for *Figure 4* and *Figure 4—figure supplement 1*.

**Figure supplement 1.** The position of the proximal border of the progenitor zone does not correlate with the position of Sh1 when data are shown in cell diameters.

### Resolving apparent inconsistencies in the literature

A recent challenge (*Gordon et al., 2020*) to previous literature (primarily *Hall et al., 1999*; *Starich et al., 2014*) was made regarding the anatomical position of Sh1. Results presented here regarding Sh1 position concur with all these published observations in a variety of genetic scenarios and provide an explanation for apparent discrepancies, one that is consistent with all of the observations to date. Our results also refute the model proposed in *Gordon et al., 2020* concerning the role of Sh1 in cell fate decisions, but reveal a novel role for innexins in cell positioning in this system. As requested by *eLife*, we include in the discussion below comments on the *Li et al., 2022* manuscript as it appeared in *bioRxiv* July 18, 2022. The major points of resolution and remaining discrepancies concern (1) the normal anatomical position of Sh1, (2) the utility of a CED-1::GFP transgene to mark Sh1, (3) the implications of Sh1 position for the stem cell fate decision, and (4) brood size analyses.

### Abnormal and normal anatomical positions of Sh1 and effects on bare regions of the distal gonad

Our results indicate that *inx-8(qy78[mKate2::inx-8])* is a dominant antimorph that significantly mispositions Sh1 to the distal end of the gonad. The mispositioning phenotype is dominant to *inx-8(+)* and the antimorphic character is revealed by our dosage analysis (*Figure 2—figure supplement 2*). As is often the case for antimorphs, residual *inx-8* function is present, consistent with the selection for *inx-8* function included in the construction strategy used to generate *inx-8(qy78[mKate2::inx-8])* (*Gordon et al., 2020*). We observed that *inx-8(0)/+* has a very mild haploinsufficient phenotype, milder than *inx-8(qy78[mKate2::inx-8])/+*. The fusion protein is therefore somewhat functional, but compromises or, in genetical terms, 'poisons' the channels into which it is incorporated.

Our finding that mKate2::INX-8 is a dominant antimorph resolves an apparent discrepancy concerning the anatomical position of the distal border of the Sh1 pair of cells as reported (*Hall et al., 1999*; *Killian and Hubbard, 2005*; *Starich et al., 2014*) and later challenged (*Gordon et al., 2020*; *Li et al., 2022*). The 2020 report demonstrated a lack of bare regions between the DTC and Sh1 in strains expressing an mKate2::INX-8 fusion protein. We now show that this marker perturbs innexin function. As a result of this perturbation, bare regions are not observed in strains carrying this fusion. Use of mKate2::INX-8 as a maker for Sh1 led to a model whereby Sh1 influences the stem cell fate decision in the underlying germ cells, participating in a direct handoff of cells from the DTC to Sh1. Instead, we show that the mKate2::INX-8 fusion itself, although it was generated by CRISPR/Cas9 genome editing in the context of the endogenous locus, mispositions Sh1 distally, obliterating the

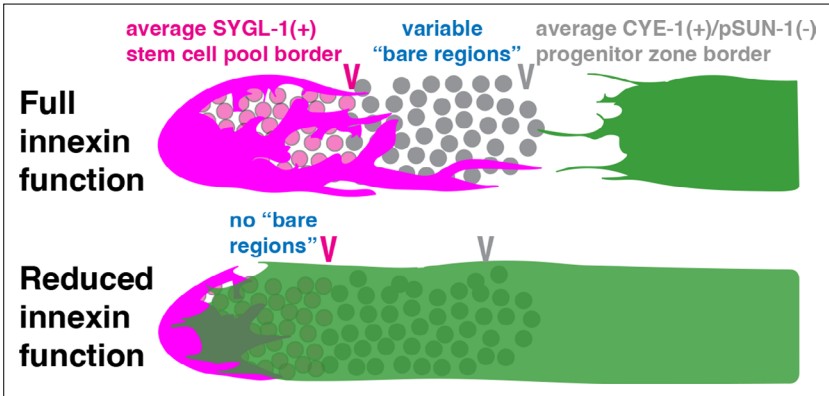

**Figure 5.** Reduced innexin function shifts Sh1 distally, eliminating the bare regions between the DTC and Sh1, but this shift does not alter the border of the stem cell pool.

bare regions. Because of the redundant function of INX-9, we were able to remove the deleterious mKate2::INX-8 protein entirely and effectively restore the bare regions (*Figure 2A, D-F*).

Notably, we also examined Sh1 position in wild-type hermaphrodites that lack fluorescent marker proteins. We confirm the presence of bare regions between the DTC and Sh1 observed with TEM (*Hall et al., 1999*), using a different fixation and staining method, high-pressure freezing/freeze-substitution (*Figure 1—figure supplement 1*, *Videos 4–5*). We also confirm the presence of bare regions observed with anti-INX-8 staining (*Starich et al., 2014*), using a different fixation and staining method, with a different imaging modality, and performed in a different laboratory (*Figure 2*). Furthermore, we show with combined anti-INX-8 staining and *inx-8(qy78[mKate2::inx-8])* mKate2 fluorescence, which fully overlap, that *inx-8(qy78[mKate2::inx-8])* results in Sh1 being distally displaced (*Figure 2—figure supplement 1*), compared to anti-INX-8 staining in the wild type (*Figure 2*). These results are fully consistent with analysis of multiple additional fluorescent transgene markers (*Figure 1*, *Figure 1—figure supplement 1*; *Figure 2*).

The cautionary tale is that fusion proteins used as markers, even when they are generated by genome editing of the endogenous locus—and therefore are not likely mis-expressed or overexpressed—may nevertheless generate proteins that are not benign. Here, the mKate2::INX-8 fusion protein caused a distal shift in the Sh1 position due to its apparent antimorphic effect on gap junctions.

## The utility of the CED-1::GFP transgene to mark Sh1

*Li et al., 2022* question the usefulness of *bcIs39[lim-7p::ced-1::gfp]*, which encodes a functional CED-1::GFP fusion as a marker for the distal edge of Sh1, showing that it has variable expression. Specifically, the hypothesis is that *inx-8(qy78[mKate2::inx-8])* illuminates distal portions of Sh1 that are not always visible with *bcIs39[lim-7p::ced-1::GFP]*. We had considered this possibility and examined strains carrying both *inx-8(qy78[mKate2::inx-8])* and either *bcIs39[lim-7p::ced-1::GFP]* or *tnIs5[lim-7p::gfp]*. We found that both GFP markers (*bcIs39* or *tnIs5*) are indeed variably expressed (faint or undetectable in 15–20% of gonads; *Figure 2F*; *Figure 2—figure supplement 3A–C*), but that this variability is not biologically meaningful with respect to the range of Sh1 position in *inx-8(+)* versus *inx-8(qy78[mKate2::inx-8])*. In the remaining 80–85% of gonads examined, the overlap between these GFP markers and the mKate2::INX-8 is complete.

Importantly, if the fusion protein encoded by *bcIs39[lim-7p::ced-1::GFP]* were not frequently illuminating the distal edge of Sh1, we would not have detected, on average, a markedly distally-displaced border of Sh1 in the *inx-14(ag17)* or '*inx-8(rf)*' mutants, nor in strains bearing the mKate2::INX-8 fusion (*Figure 1C–G*; *Figure 1—figure supplement 1B–E*; *Figure 2D–F*; *Figure 2—figure supplement 2B–F*). That we reproducibly detect a significantly distally displaced border of Sh1 with these GFP markers in strains with aberrant innexin function, as well as with an additional independent Sh1 marker (*Figure 1—figure supplement 1B–E*), further suggests that they reliably mark the entire cell. *Li et al., 2022* also present an over-exposed and saturated image of the distal gonad without providing a control strain lacking the *bcIs39* marker. Thus, it remains uncertain whether what is seen in this image relates to the marker.

One concern raised by *Li et al., 2022* were morphological defects they observed in 22% of gonads of worms expressing a CED-1::GFP fusion protein. The remaining 78% of gonads exhibit a 'gap' or an overlapping 'interface' with the DTC, categories not unlike those we classified within Class 1 and Class 2. Because our analysis focused on those live worms that were oriented in such a way to image the entire distal gonad, it is possible that a minority of gonads were censored due to gonad migration defects. For example, it may not have been possible to distinguish gonad arms with migration defects from those where the gonad was obscured by the intestine. That said, when we imaged DG5133 *inx-8(qy102) inx-9(ok1502)*, it was immediately obvious that many gonad arms were grossly misshapen, so we are confident in our ability to detect penetrant gross morphological defects. Like *Li et al., 2022*, we observe that one of the two Sh1 cells is occasionally positioned slightly more proximal than the other. This is likely the result of a birth order difference of their ancestor sheath/spermatheca precursor cells during somatic gonad development (*Kimble and Hirsh, 1979*).

More salient than the penetrance of migration defects in strain DG5020 bearing *naIs37* and *bcIs39*, is that all of the Sh1 markers we used (including *bcIs39*, *tnIs5*, *tnIs6*, *fasn-1*(av138), and *acy-4(tnEx42)*; *Figure 1—figure supplement 1*; *Figure 2*) show a similar range of Sh1 positioning (represented by Class 1 and Class 2) and that this position matches the previous and current TEM analyses (*Hall*

*et al., 1999*; this work *Figure 1—figure supplement 1*, *Videos 4–5*) as well as the anti-INX-8 analysis (*Figure 2*; *Figure 2—figure supplement 1*). *Li et al., 2022* also incorrectly state that Sh1 was thought to associate only with germ cells well into meiotic development. However, a previously published time course analysis demonstrates association of Sh1 with the progenitor zone during development (see *Killian and Hubbard, 2005*, *Figures 2 and 3*).

An unresolved question is why several membrane-associated Sh1 markers exhibit patterns that are discordant with strains bearing *bcls39, tnls5, tnls6, fasn-1(av138), acy-4(tnEx42)* markers and with TEM and antibody analyses in unmarked strains. We hypothesize that, like strains bearing an mKate::INX-8 fusion protein, membrane-associated fluorescent proteins described by *Li et al., 2022* similarly disrupt functions that act to position Sh1, including CAM-1::mNeonGreen, INA-1::mNeonGreen, and *lim-7p*::GFP::CAAX that display reduced or absent bare regions. Since the gonadal architecture displayed by these markers differs from that established by analysis of unmarked strains (*Hall et al., 1999*; *Starich et al., 2014*; this work), we hypothesize that these proteins, like mKate2::INX-8, disrupt the physiology of Sh1 in some manner. If so, they may, for example, display reduction-of-function or even dominant effects on the position of Sh1, as do strains bearing mKate2::INX-8. Although such tests of this hypothesis are beyond the scope of our study, it will be interesting to determine how these fusion proteins are perturbing Sh1 position and whether these membrane markers perturb Sh1–germline gap junctions.

## The implications of Sh1 position for the stem cell fate decision

*Li et al., 2022* experimentally altered germ cell fate or proliferative capacity of germ cells and assayed Sh1 position, whereas we tested directly the proposed model that Sh1 influences the stem cell border (*Gordon et al., 2020*) by altering Sh1 position and assaying expression of the direct GLP-1 target SYGL-1. *Li et al., 2022* report that the boundary of Sh1 correlates with the group of germ cells that is first to differentiate when GLP-1/Notch signaling or mitosis are blocked using temperature-sensitive mutations in *glp-1* or *emb*-30, which respectively encode the GLP-1/Notch receptor (*Austin and Kimble, 1989*; *Yochem and Greenwald, 1989*) or the APC4 subunit of the Anaphase-Promoting Complex (*Furuta et al., 2000*). However, in all these experiments, Sh1 was marked with the mKate2::INX-8 fusion, which we demonstrate here acts as a dominant antimorph that causes distal mispositioning of Sh1. While the mispositioned Sh1 border in strains expressing mKate2::INX-8 does indeed correlate with the SYGL-1 border, if Sh1 were influencing the stem cell fate decision as proposed, the SYGL-1 border should move with Sh1, and it does not (*Figure 3*).

To assess the stem cell border, we examined GLP-1/Notch signaling using a direct target readout, SYGL-1 (*Chen et al., 2020*; *Kershner et al., 2014*; *Lee et al., 2019*; *Lee et al., 2016*; *Shin et al., 2017*). *Li et al., 2022*, use comparatively indirect proxies. These include the presence of cells in mitotic prophase, which do not distinguish mitotic stem cells from mitotic non-stem progenitors (*Fox and Schedl, 2015*), and the position of overt entry into meiotic prophase. The latter indicates the border of the progenitor zone (PZ) that includes mitotically competent non-stem progenitors (see *Hubbard and Schedl, 2019* for a review of the germline stem cell system). We had also examined the relationship between Sh1 position and the PZ border as indicated with anti-CYE-1 antibody (*Figure 4*), and we concur that the progenitor zone is modestly but significantly shorter in strains bearing mKate2::INX-8 compared to strains that are otherwise similarly marked (*Figure 4*). This is not surprising since *inx-8(qy78[mKate2::inx-8]*) encodes a dominant anti-morph and the *inx-8 inx-9* double mutant has a profound effect on germ cell proliferation, though not on cell fate as mediated by *glp-1* (*Starich et al., 2014*). Nonetheless, the observed effect on the extent of the PZ is very modest compared to the effect of this allele on the position of Sh1, and on a gonad-by-gonad basis there is no correlation (*Figure 4B–C*).

Our finding that the position of the germline response to signaling from the DTC, as measured by expression of the direct GLP-1/Notch target, SYGL-1, is independent of the position of Sh1 (*Figure 3*) is also consistent with other observations from the literature. For example, males, which lack sheath cells altogether, exhibit stem cell pools similar to those in the hermaphrodite (*Crittenden et al., 2019*). In addition, the stem cell and progenitor pools are present in larval stages prior to DTC process elaboration (*Byrd et al., 2014*), when Sh1 is positioned several CD from the DTC, precluding the possibility of a DTC–Sh1 handoff at these stages (e.g. L4 stage, when stem/non-stem fate decisions are in progress; *Killian and Hubbard, 2005*). Here, we observed that the distal border of Sh1 relative to the proximal stem cell border in the wild type is ≥5 CD in the majority of adult gonads we examined. Thus, the model suggested in *Gordon et al., 2020*, that Sh1 physically orients divisions of stem cells

to direct their fate, is also called into question by our results. Finally, using alleles that dramatically alter the position of Sh1, we found no evidence supporting the prediction that the stem/non-stem border is coincident with the Sh1 border. Together, these results indicate that Sh1 is not involved in the germline stem-progenitor fate decision.

### Brood size analyses

An unresolved difference between our study and that of *Li et al., 2022* concerns observations of brood sizes in strains bearing *bcIs39[lim-7p::ced-1::gfp]*. In our studies, a strain bearing *bcIs39* as the only marker (DG5380, *Table 1*), exhibited a brood size (282.6±34.2, n=25), which was not statistically different (p=0.1501, *t* test) from that of the wild type (298.6±42.1, n=24). *Li et al., 2022* did not analyze this exact strain, they did analyze DG5020 *naIs37*[*lag-2p::mCherry-PH*]; *bcIs39[lim-7p::ced-1::gfp]* and observed ~20% embryonic lethality, while we only observed ~1% embryonic lethality (*Table 1*). This difference might reflect variations in growth conditions despite the fact that both groups utilized standard *C. elegans* culture conditions. Perhaps relevant, and as noted in the results section above, key germline metabolites such as malonyl Co-A require gap junctions and innexin function for transport (*Starich et al., 2020*). Alternatively, such a difference might be due to genetic drift.

### The role of bare regions in the adult hermaphrodite gonad is uncertain

Our observation that loss of the bare regions correlates with only a relatively modest effect on brood size (*Table 1*) raises the question of what, if any, biological significance the bare regions might have. Because the innexins have multiple functions in germline development (*Starich et al., 2014*; *Starich et al., 2020*; *Starich and Greenstein, 2020*), we are unable to ascribe the observed brood-size reduction in strains lacking bare regions to the absence of the bare regions per se. Thus, any potential roles for the bare regions vis-à-vis brood size and germline development will require additional study.

### Gap junctions position somatic cells in distal gonad

Our studies also provide evidence that innexin gap junctions not only serve as communication and adhesion junctions, but that in the context of an organ system, they contribute to the positioning of cells relative to each other. How might gap junctions influence the relative position of the DTC and Sh1 in the distal gonad arm? The DTC forms gap junctions with germ cells, which must be disassembled as germ cells enter a bare region, only to be reassembled again when in contact with Sh1 (and then again with more proximally located sheath cells Sh2–5). A detailed TEM analysis of the gonad (*Hall et al., 1999*) led to the consideration that a constant interplay of association and dissociation likely occurs between Sh1 and the underlying germ cells that move proximally along the arm: as germ cell flux continually moves germ cells towards the proximal end of the gonad, the Sh1 cells presumably extend their filopodia distally and form new gap junction connections with incoming germ cells. Otherwise, the bare regions would increase in length along the distal-proximal axis. Under optimal growth conditions in the laboratory, this constant association and dissociation between Sh1 and germ cells may be pushed to its limit.

To complement the role of gap junctions in promoting robust proliferation revealed by the *inx-8 inx-9* double loss-of-function mutant or loss of the germline-encoded channel components, the kinetics of gap junction coupling between the somatic gonad and germ cells may play a role in determining the strength of the interactions between the two cell types. Unlike sheath–oocyte junctions, which form large plaques containing many functional gap junctions, the gap junctions formed in the distal arm appear to represent looser associations of a few gap-junction channels (*Starich et al., 2014*). Nonetheless, these associations may be sufficient to maintain adhesion with the underlying germline, functioning like regularly spaced rivets, albeit dynamic and removeable ones.

Disentangling the adhesive and channel functions of gap junctions is a complex issue. The mutants used in this study are competent to form soma–germline gap junctions—without these, germ cells fail to proliferate. However, they may form such junctions less efficiently than their wild-type counterparts. For example, the pattern of localization of gap junction puncta in *inx-8(rf)* and *inx-14(ag17)* appears more diffuse than in the wild type (*Starich et al., 2014*; *Starich et al., 2020*). Alternatively or additionally, the mutant innexins in this study may assemble into hemichannels as readily as wild-type innexins, but the pairing or opening of gap junction channels may be compromised. Studies of connexin gap-junction channels in paired *Xenopus* oocytes strongly suggest that opening of hemichannels facilitates

their assembly into gap junctions. That study proposed hemichannel opening collapses the intermembrane space between juxtaposed cells to allow the extracellular loops of connexins to dock into gap junctions (*Beahm and Hall, 2004*). If a similar model applies to innexin-containing gap junctions, then rivet and channel function would be coupled.

How could impaired innexin function cause Sh1 to creep distally? One hypothesis is that when fewer junctions are made, Sh1 cannot adhere as well to germ cells. This scenario would be consistent with the observation that Sh1 is distally mispositioned when *inx-8* and *inx-9* are missing from Sh1 but *inx-8(+)* is present in the DTC by virtue of heterologous expression of *inx-8(+)* in the DTC in an *inx-8(0) inx-9(0)* double mutant (*Starich et al., 2014*). It is also possible that the DTC and Sh1 engage in an active repellent or a passive space-excluding interaction that somehow involves gap junction function. Another possibility is that a deficit in gap-junction function might trigger Sh1 spreading as a means to increase the surface area over which junctions may form to supply sufficient active biomolecules that transit through these junctions. Nevertheless, our studies show that the position of the germline stem cell decision is not dictated by the position of Sh1.

## Materials and methods
### Live imaging and image analysis of live samples
Live specimens were grown at 20 °C, and staged by picking mid-L4 larvae, then allowing them to grow at 20 °C until imaging them 24 hr later. Animals were immobilized using 10 mM Levamisole (Sigma T1512) in M9 buffer. Imaging was carried out on a Nikon W1 spinning disk confocal microscope. We define a biological replicate as any set of individuals collected, processed and imaged together. Numbers of individuals included in each analysis is indicated in the figure panels or legends.

Image analysis was carried out on two-dimensional maximum-projection Z-stack images of 3D confocal data. The distance from the distal end of the gonad to the end of each DTC process was measured along a line drawn from the end of each process parallel to the distal-proximal axis to a line drawn tangent to the distal end, orthogonal to the distal-proximal axis line (*Figure 1—figure supplement 1*). The distance between the distal end of the gonad and the most distal extent of Sh1 was measured in the same way. All data points were recorded for each sample and used to calculate the mean values presented in *Figures 1 and 2* and related figure supplements.

Sh1 visualization in live worms: *bcIs39* [*lim-7p*::CED-1::GFP] (*Zhou et al., 2001*) encodes a functional membrane-localized fusion to CED-1. *tnIs5* and *tnIs6* encode an identical non-functional fusion to the first 61 amino acids of LIM-7 (*tnIs5* or *tnIs6*) denoted here as '*lim-7p*::GFP' that includes 2.23 kb upstream, the first two exons, and the first intron of *lim-7* fused to GFP (*Hall et al., 1999*). *inx-8(qy78[mKate2::inx-8])* is an N-terminal fusion of mKate2 to INX-8 generated by CRISPR-Cas9 genome editing (*Gordon et al., 2020*). *fasn-1(av138[fasn-1::gfp])* was generated by fusing GFP to the C-terminus of FASN-1 using CRISPR-Cas9 genome editing (*Starich et al., 2020*). *tnEx42[acy-4::gfp +rol-6(su1006)]* contains GFP fused to the ACY-4 C-terminus by recombineering in the fosmid WRM061bE10 (*Govindan et al., 2009*). ACY-4::GFP rescues the sterility of *acy-4(ok1806)* null mutants via its function in the gonadal sheath cells.

DTC visualization in live worms: *naIs37*[*lag-2p*::mCherry-PH] encodes mCherry fused to the PH domain of rat phospholipase C delta (*Pekar et al., 2017*) and *qIs154*[*lag-2p*::MYR-tdTomato] encodes a src kinase myristoylation sequence fused to tdTomato (*Byrd et al., 2014*).

### Imaris rendering of confocal images
3D rendering of confocal stacks for *Videos 1–3* was done using Imaris (Oxford Instruments, plc. Abingdon, UK) according to the following algorithm for batch processing. For the Sh1 channel (green): Enable Smooth = true; Surface Grain Size = 0.433 µm; Enable Eliminate Background = false; Diameter Of Largest Sphere = 1.63 µm; Enable Automatic Threshold = false; Manual Threshold Value = 650; Active Threshold = true; Enable Automatic Threshold B=true; Manual Threshold Value B=31622.8. For the DTC channel (red): Enable Smooth = true; Surface Grain Size = 0.433 µm; Enable Eliminate Background = false; Diameter Of Largest Sphere = 1.63 µm; Enable Automatic Threshold = false; Manual Threshold Value = 710; Active Threshold = true; Enable Automatic Threshold B=true; Manual Threshold Value B=14988.5.

## Strains

*C. elegans* strains (*Supplementary file 1*) were grown on standard NGM media with OP50 as a food source (*Brenner, 1974*) for all figures in this work. Strain constructions employed a modified NGM medium [containing 6.25 μg/ml Nystatin and 12.5 mM potassium phosphate pH 6.0 (added after autoclaving) and 200 mg/ml streptomycin sulphate (added before autoclaving)] with *E. coli* strain OP50-1 as food source. Strains were grown at 20 °C. In addition to the wild-type strain N2, the following alleles, described in WormBase (https://wormbase.org//#012-34-5) or in the cited references, were used:

Chr. I—*dpy-5(e61)* (*Brenner, 1974*), *inx-14(ag17)* (*Miyata et al., 2008*; *Starich et al., 2014*), *unc-13(e51)* (*Brenner, 1974*), *fasn-1(av138[fasn-1::gfp])* (*Starich et al., 2020*), *sygl-1(q983[3xOLLAS::sygl-1])* (*Shin et al., 2017*), *unc-54(e190)* (*Brenner, 1974*).

Chr. IV—*inx-8(qy78[mKate2::inx-8])* (*Gordon et al., 2020*), *inx-8(qy78 tn2031)* (this work), *inx-8(tn2034)* (this work), *inx-8(tn2075)* (this allele was generated on the *tmC5* balancer chromosome, this work), *inx-9(ok1502)*, *inx-8(qy102[mKate2::inx-8]) inx-9(ok1502)* (*Gordon et al., 2020*), *inx-8(tn1513tn1553) inx-9(ok1502)* (*Starich and Greenstein, 2020*), *inx-8(tn1513tn1555) inx-9(ok1502)* (*Starich et al., 2020*), *dpy-20(e1282)* (*Hosono et al., 1982*).

Chr. V—*acy-4(ok1806)* (*Govindan et al., 2009*).

Balancer chromosomes *Dejima et al., 2018* used included: *tmC18 [dpy-5(tmIs1236)]* I, *tmC27[tmIs1239]* I, *tmC5[tmIs1220]* IV.

Integrated transgenes included: *naIs37[lag-2p::mCherry::plcdeltaPH +unc-119(+)]* I (*Pekar et al., 2017*), *mIs11[myo-2p::gfp +pes-10p::gfp +gut promoter::gfp]* IV, *bcIs39[lim-7p::ced-1::gfp +lin-15(+)]* V (*Zhou et al., 2001*), *qIs154[lag-2p::MYR::tdTomato +ttx-3p::gfp]* V (*Byrd et al., 2014*), *tnIs5[lim-7p::gfp +rol-6(su1006)]* X, *tnIs6[lim-7p::gfp +rol-6(su1006)]* X (*Hall et al., 1999*), *cpIs122[lag-2p::mNeonGreen::plcdeltaPH]* (*Gordon et al., 2020*).

Extrachromosomal arrays used included: *tnEx42[acy-4::gfp +rol-6(su1006)]* (*Govindan et al., 2009*).

## Strain constructions

Multiply mutant strains were constructed in a straightforward manner (*Huang and Sternberg, 1995*). *tmC18* was used as a balancer chromosome for *inx-14(ag17)*. *tmC27* was used as a balancer chromosome for *sygl-1(q983)*. *tmC5* (or the derivatives described) or *mIs11* were used as balancer chromosomes for *inx-8* and *inx-9* mutant alleles. The presence of *inx-14(ag17)* in strains was verified by PCR and DNA sequencing. The *ag17* allele was originally described as an Arg to His change in the second extracellular loop of INX-14, but the exact residue position was not specified (*Miyata et al., 2008*). A 1.2 kb PCR fragment covering this region was amplified with primers inx-14delF and inx-14delR (see *Supplementary file 2* for the sequence of oligonucleotides used in this study). The PCR fragment was sequenced with the inx-14delR primer. No sequence changes were found in Arg residues predicted to occupy the second extracellular loop. However, a CGT to CAT (R326H) change was identified at a residue position predicted to lie near the cytoplasmic end of the fourth transmembrane domain, and we surmise that this change represents the original *ag17* mutation. The presence of the *sygl-1(q983[3xOLLAS::sygl-1])* mutation in strains was verified by PCR with primers sygl1-F and sygl1-R, which produce a 216 bp product in the wild type and a 348 bp product in *sygl-1(q983)* and by anti-OLLAS staining. The presence of *inx-8(qy78tn2031)* and *inx-8(tn2034)* in strains was verified by PCR with oligonucleotide primers inx8_delta.F and inx8_delta.R. To construct DG5367 *inx-14(ag17) fasn-1(av138[fasn-1::gfp]) naIs37 I, fasn-1(av138[fasn-1::gfp]) naIs37/dpy-5(e61) unc-13(e51)* heterozygotes were generated and Dpy non-Unc recombinants were sought, which resulted in the isolation of a *dpy-5(e61) fasn-1(av138[fasn-1::gfp]) naIs37 I* intermediate strain. From, *inx-14(ag17)/dpy-5(e61) fasn-1(av138[fasn-1::gfp]) naIs37* heterozygotes, recombinants of genotype *inx-14(ag17) fasn-1(av138[fasn-1::gfp]) naIs37/dpy-5(e61) fasn-1(av138[fasn-1::gfp]) naIs37* were identified by virtue of their increased GFP signal using a fluorescence dissecting microscope. The presence of *inx-14(ag17)* in the desired recombinants was verified by DNA sequencing. *naIs37* was shown to be tightly linked to *unc-54(e190)* on the right end of LGI as follows. From *dpy-5(e61) fasn-1(av138[fasn-1::gfp]) naIs37/unc-54(e190)* heterozygotes, of 15 *unc-54(e190)* homozygotes, none contained the *naIs37* marker (4 were recombinants expressing *fasn-1::gfp*). Of 38 wild-type segregants of the aforementioned parental heterozygous strain, all expressed the *naIs37* marker (11 were recombinants lacking *fasn-1::gfp* expression). DG5380 was derived from DG5020 by crossing to N2 males and segregating away *naIs37*; the *bcIs39* source was MD701, available from the CGC.

## Electron microscopy methods and analysis

Wild-type N2 young adults were analyzed using high-pressure freezing/freeze-substitution freezing (HPF/FS; *Hall et al., 2012*). Four animals from chemical immersion and four animals from HPF/FS were collected in serial sections on slot grids, ranging from 80 nm to 100 nm thickness on an RMC PowerTome XL ultramicrotome (Eden Instruments, Valence, France). Sections were post-stained in 2% uranyl acetate in $H_2O$ for 20 min and in Reynold's lead for 3 min. Sections from each animal were viewed in the gonad region using Digital Micrograph software (Gatan, Pleasanton, CA) for the JEOL JEM-1400Plus M (Jeol USA, Peabody, MA), using a Gatan Orius SC1000B digital camera.

We selected the most optimally positioned and resolved HPF/FS-fixed animal (2% osmium tetroxide, 0.1% uranyl acetate, and 2% $H_2O$ in acetone) to collect high-resolution montages of the gonad arm from every third section covering 80 µm, including the DTC and sheath filopodia of a posterior distal gonad arm. This series was collected from the tail to the vulva region to encompass the whole posterior gonad arm, but we did not collect full images closer to the gonad reflection in order to save effort and expense. The pixel size for the images was 3.23 nm. The images were aligned with TrakEM2 software (*Cardona et al., 2012*) and traced on a Wacom DTZ 2100D tablet (Wacom, Portland, OR) to build the 3D model.

We followed the DTC processes and their fragments and the filopodial extensions of Sh1 due to their increased translucency compared with the germ cells and by the fact that these somatic cellular extensions contained many small mitochondria (*Video 4*). We observed what we infer to be shed pieces of the DTC beginning at approximately 30 µm from the distal end, as has been observed using light microscopic methods (*Byrd et al., 2014*). However, some DTC and Sh1 processes may appear discontinuous due to the fact that micrographs of every third section were used to generate the reconstructed model (*Video 5*). Regardless of the effect that this sampling technique has on our ability to resolve contiguous versus shed bits of somatic gonadal cells, this TEM analysis is in agreement with our light microscopic observations that a large subset of distal germ cell progenitors are not in contact either with the DTC or Sh1.

## Brood counts and embryonic lethality measurements

Because *inx-8(qy78[mKate2::inx-8])* brood counts and embryonic lethality were highly variable, we used a single batch of NGM media with OP50 food source for all brood count measurements reported in *Table 1*. To prevent mold formation, the medium contained 6.25 µg/ml Nystatin. L4-stage hermaphrodites were cultured individually on 35 mm Petri plates and transferred approximately every 24 hr until they stopped producing embryos (4–6 days). Worms that crawled off the media and died were redacted (varied from 0 to 10% depending on the experiment). Embryos that failed to hatch after 24–36 hr were counted and scored as dead. In the majority of cases, these embryos exhibited morphological abnormalities. Control experiments demonstrated that these embryos were not simply delayed and never hatched. Embryos that hatched were counted and scored as viable.

## Genome editing

CRISPR-Cas9 genome editing was used to generate *inx-8* null alleles in both the *inx-8(qy78[mKate2::inx-8])* and wild-type genetic backgrounds. The approach taken generated identical 1525 bp deletions within the *inx-8* locus in both genetic backgrounds starting 136 bp upstream of the wild-type *inx-8* ATG start codon and extending 221 bp into *inx-8* exon 3. In the *inx-8(qy78[mKate2::inx-8])* context, this edit removes both the mKate2 moiety and *inx-8*. The deletions are expected to constitute *inx-8* null alleles because, in addition to removing the start codon, they delete amino acids 1–349 (out of 382 amino acids), including virtually all residues essential for spanning the plasma membrane and forming a channel (*Starich and Greenstein, 2020*). The approach used pRB1017 to express two single guide RNAs (sgRNAs) under control of the *C. elegans* U6 promoter (*Arribere et al., 2014*). Oligonucleotides inx8_us_sgRNA1.F and inx8_us_sgRNA1.R were annealed and used to generate the plasmid inx8_us_sgRNA1 to direct Cas9 cleavage 136 bp upstream of the ATG initiator codon (*Supplementary file 2* lists all oligonucleotides used in this study). Oligonucleotides inx8_sgRNA1.F and inx8_sgRNA1.R were annealed and used to generate the plasmid inx8_sgRNA1 to direct Cas9 cleavage in exon 3. To generate sgRNA clones, annealed oligo-nucleotides were ligated to *Bsa*I-digested pRB1017 plasmid vector, and the resulting plasmids were verified by Sanger sequencing. pDD162 served as the source of Cas9 expressed under control of the *eef-1A.1/eft-3* promoter (*Dickinson et al., 2013*). The repair template oligonucleotide used was inx8_rpr.

Genome editing employed the *dpy-10* co-conversion method (*Arribere et al., 2014*). The injection mix contained pJA58 (7.5 ng/µl), AF-ZF-827 (500 nM), inx8_us sgRNA1 (25 ng/µl), inx8_sgRNA1 (25 ng/µl), inx8_rpr (500 nM), and pDD162 (50 ng/µl) and was injected into adult hermaphrodites from strains DG5131 *naIs37[lag-2p::mCherry::PH +unc-119(+)] I; inx-8(qy78[mKate2::inx-8]) IV; bcIs39[lim-7p::ced-1::gfp +lin-15(+)] V* and DG5020 *naIs37[plag-2::mCherryPH +unc-119(+)] I; bcIs39[lim-7p::ced-1::gfp +lin-15(+)]V*. Correct targeting was verified by conducting PCR with primer pairs inx8_delta.F and inx8_delta.R followed by DNA sequencing. Three deletion alleles were recovered from the injections into DG5131 (*qy78tn2031*, *qy78tn2032*, and *qy78tn2033*), and two deletion alleles were recovered from the injections into DG5020 (*tn2034* and *tn2035*). The deletion alleles were outcrossed to *naIs37[lag-2p::mCherryPH +unc-119(+)]/+I; tmC5(tmIs1220[myo-2p::Venus])/+IV; bcIs39[lim-7p::ced-1::gfp +lin-15(+)]/+V* males to generate DG5229 and DG5232. Homozygous strains were analyzed by confocal microscopy. To introduce an *inx-8* null allele onto the *tmC5[tmIs1220] IV* balancer chromosome, the same injection mix of CRISPR reagents as described above was injected into *dpy-20(e1282)/tmC5[tmIs1220] IV* heterozygotes. The *inx-8(tn2075)* null allele was generated on the balancer chromosome, which resulted from a 1526 bp deletion that differed from the other CRISPR-generated *inx-8* null alleles by the removal of an extra base at the cut site.

## Immunostaining and image analysis of fixed samples

Immunostaining was carried out as described (*Mohammad et al., 2018*). Briefly, synchronized adult hermaphrodites, 24 hr past mid-L4, were dissected in PBST (PBS with 0.1% Tween 20), with 0.2 mM levamisole to extrude the gonads. The gonads were fixed in 3% paraformaldehyde solution for 10 min and then post-fixed in −20° chilled methanol for 10 min. After 3x10 min washes with PBST, they were blocked in 30% goat serum for 30 min at RT. The gonads were then incubated with the desired primary antibodies diluted (see below) in 30% goat serum at 4° overnight. The next day, after 3x10 min PBST washes, the gonads were further incubated with appropriate secondary antibodies, diluted in 30% goat serum, at 4° overnight. The gonads were washed 3 times with PBST, then incubated with 0.1 g/ml DAPI in PBST for 30 min. After removal of excess liquid, the gonads were mixed with anti-fading agent (Vectashield) and transferred to an agarose pad on a slide. Hyperstack images were captured using a spinning disk confocal microscope (PerkinElmer-Cetus, Norwalk, CT). Two overlapping hyper-stack images were captured for each gonad arm to obtain coverage of >50 cell diameters from the distal end of the gonad. Images were further processed in Fiji, and DAPI stained nuclei were used to mark the number of cell diameters from the distal end. Employing pixel to micron ratio, specific to the images captured, cell diameters were converted into microns where required.

### SYGL-1-positive zone length assessment

OLLAS staining was used to assess 3xOLLAS::SYGL-1 accumulation (*Shin et al., 2017*). In wild-type young adults, SYGL-1 accumulates at the distal end of the germline and is downregulated around 10-cell diameters from the distal tip (*Kocsisova et al., 2019*; *Shin et al., 2017*). Cell diameters were counted from the distal end of the germline up to the row where SYGL-1 is no longer visible by eye. OLLAS staining in the wild-type worms without OLLAS tag was used to differentiate staining from the background. To confirm the accuracy of our visual assessment, we quantified the intensity of SYGL-1 accumulation in the distal germline, employing methods similar to *Chen et al., 2020* in the same set of germlines where the SYGL-1 zone was visually evaluated. We found that the cell diameter position called as the end of the SYGL-1 zone consistently corresponded to 6–9% of peak SYGL-1 intensity, for each genotype (*Figure 3—figure supplement 2*). These results indicate that the SYGL-1 zone length visual assessment was reproducible and consistent.

### Progenitor zone length assessment

The gonads were stained with a progenitor zone marker, anti-CYE-1, and an early meiotic prophase marker, pSUN-1, (anti-SUN-1 S8-Pi) (*Mohammad et al., 2018*). For assessing the progenitor zone length, cell diameters (rows) were counted from the distal end of the germline, where all cells are CYE-1-positive, until the point after which the majority of the cells in a row have switched from staining for CYE-1 to pSUN-1. Note that pSUN-1 staining is not shown in the figures, although it was used to assess the PZ border.

## Assessment of distal position of Sh1

Anti-GFP antibody staining was used to visualize the sheath, where cell diameters were counted from distal end to the point where GFP staining became prominent.

To quantify the distance from the distal end to the border of expression of any protein, we have used either ~30 gonads for single-replicate experiments and ~20 gonads for multi-replicate experiments. We define a biological replicate as independently dissected and stained, along with the controls in the same tube, and images acquired with the same settings. During these experiments, no outliers were encountered. In rare cases, acquired images, which either failed to stitch by Fiji or had poor staining, were discarded.

## Primary antibodies used

mouse anti-CYE-1 (1:100; *Brodigan et al., 2003*); guinea pig anti-SUN-1 S8-Pi (1:1000; *Penkner et al., 2009*); rat anti-OLLAS (1:2000; Novus Biological); rabbit anti-GFP (1:200; from Swathi Arur, MD Anderson Cancer Center; *Lopez et al., 2013*); and rabbit anti-INX-8 (1:25; *Starich et al., 2014*).

## Secondary antibodies used

Alexa Fluor 647 goat anti-mouse (Life Technologies, Carlsbad, CA); Alexa Fluor 594 goat anti-guinea pig (Invitrogen, Waltham, MA), Alexa Fluor 594 donkey anti-rat (Invitrogen); and Alexa Fluor 488 goat anti-rabbit (Invitrogen).

## Acknowledgements

We thank Gabriela Huelgas-Morales for discussions and constructive suggestions during the course of this work, Swathi Arur for the affinity purified anti-GFP antibody, Verena Jantsch for anti-pSUN-1 antibody, Edward Kipreos for anti-CYE-1 antibody. We thank Michael Cammer and the NYU Langone Microscopy Core for experimental and technical support; the NYU Langone Microscopy Laboratory is partially supported by the Cancer Center Support Grant P30CA016087 at the Laura and Isaac Perlmutter Cancer Center. We thank WormBase. We thank Kacy Gordon and the CGC (which is funded by NIH Office of Research Infrastructure Programs P40 OD010440), for strains. NIH P30HD071593 and 1S10OD016214-01A1 supports electron microscopy in DHH laboratory.

## Additional information

### Competing interests

E Jane Albert Hubbard: while not directly related, in the interest of full disclosure, E.J.A.H. holds US patent 6,087,153. The other authors declare that no competing interests exist.

### Funding

| Funder | Grant reference number | Author |
|---|---|---|
| American Cancer Society | PF-19-231-01-CSM | Theodora Tolkin |
| National Institutes of Health | R35GM134876 | E Jane Albert Hubbard |
| National Institutes of Health | R01AG065672 | E Jane Albert Hubbard |
| National Institutes of Health | R01GM100756 | Tim Schedl |
| National Institutes of Health | R01GM57173 | David Greenstein |
| National Institutes of Health | R35GM144029 | David Greenstein |
| National Institutes of Health | R24OD010943 | David H Hall |

| Funder | Grant reference number | Author |
|--------|------------------------|--------|

The funders had no role in study design, data collection and interpretation, or the decision to submit the work for publication.

## Author contributions

Theodora Tolkin, Conceptualization, Data curation, Formal analysis, Funding acquisition, Investigation, Visualization, Methodology, Writing – original draft, Writing – review and editing; Ariz Mohammad, Data curation, Formal analysis, Investigation, Visualization, Methodology, Writing – review and editing; Todd A Starich, Conceptualization, Formal analysis, Investigation, Methodology, Writing – review and editing; Ken CQ Nguyen, Data curation, Validation, Visualization, KCQN conducted TEM studies and 3D reconstructions; David H Hall, Conceptualization, Resources, Formal analysis, Supervision, Visualization, Methodology, Writing – review and editing, DHH designed and supervised TEM studies and the data analysis; Tim Schedl, Conceptualization, Supervision, Funding acquisition, Project administration, Writing – review and editing; E Jane Albert Hubbard, Conceptualization, Supervision, Funding acquisition, Methodology, Writing – original draft, Project administration, Writing – review and editing; David Greenstein, Conceptualization, Supervision, Funding acquisition, Investigation, Methodology, Project administration, Writing – review and editing

## Author ORCIDs

Theodora Tolkin ⬛ http://orcid.org/0000-0003-2174-3885
Ariz Mohammad ⬛ http://orcid.org/0000-0002-2807-412X
Ken CQ Nguyen ⬛ http://orcid.org/0000-0002-3479-2611
David H Hall ⬛ http://orcid.org/0000-0001-8459-9820
E Jane Albert Hubbard ⬛ http://orcid.org/0000-0001-5893-7232
David Greenstein ⬛ http://orcid.org/0000-0001-8189-2087

## Decision letter and Author response

Decision letter https://doi.org/10.7554/eLife.74955.sa1
Author response https://doi.org/10.7554/eLife.74955.sa2

# Additional files

## Supplementary files

- Supplementary file 1. Strains used in this study.
- Supplementary file 2. Oligonucleotides used in this study.
- Transparent reporting form

## Data availability

All data generated or analysed during this study are included in the manuscript and supporting files; Source Data files have been provided where appropriate.

The following dataset was generated:

| Author(s) | Year | Dataset title | Dataset URL | Database and Identifier |
|-----------|------|---------------|-------------|-------------------------|
| Tolkin T, Mohammed A, Starich T, Nguyen KCQ, Hall DH, Schedl T, Hubbard EJA, Greenstein D | 2022 | Serial thin section movie of every third section from the DTC to the distal extensions of Sh1 in a young adult hermaphrodite posterior gonad arm | https://doi.org/10.5061/dryad.xgxd254j4 | Dryad Digital Repository, 10.5061/dryad.xgxd254j4 |

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

# Appendix 1

## Appendix 1—key resources table

| Reagent type (species) or resource | Designation | Source or reference | Identifiers | Additional information |
|---|---|---|---|---|
| Antibody | mouse monoclonal anti-CYE-1 | *Brodigan et al., 2003* | | (1:100) |
| Antibody | rat monoclonal anti-OLLAS | *Shin et al., 2017* | | (1:2000) |
| Antibody | guinea pig polyclonal anti-SUN-1 S8-Pi | *Mohammad et al., 2018* | | (1:1000) |
| Antibody | rabbit polyclonal anti-GFP | from Swathi Arur, MD Anderson Cancer Center; *Lopez et al., 2013* | | (1:200) |
| Antibody | rabbit polyclonal anti-INX-8 | *Starich et al., 2014* | | (1:25) |
| sequence-based reagent | AF-ZF-827 | *Arribere et al., 2014* | | CACTTGAACTTCAATA CGGCAAGATGAGAAT GACTGGAAACCGT ACCGCATGCGGTG CCTATGGTAGCGGA GCTTCACATGGCTTCAG ACCAACAGCCTAT Contact DG for more information |
| sequence-based reagent | inx8_delta.F | this work | | CCTTCGACCTGATTT CCCCTTCTTCTAATG Contact DG for more information |
| sequence-based reagent | inx8_delta.R | this work | | CTATTGCTTTCCGTT CTTCAAGATGTTGTTG Contact DG for more information |
| sequence-based reagent | inx8_RPR | this work | | GGTGGCCAATAAA AATGCTTTTCTTTTT GCTTTTCTCTATCTA CTTCCGTTCCGCCCC GGAGGTTGCCGTGG AGATGTACAGCGAC TTTTTAGTAAGTCTT TTCAAC Contact DG for more information |
| sequence-based reagent | inx8_sgRNA1.F | this work | | TCTTGAGTGACTTGG TAGCATCGG Contact DG for more information |
| sequence-based reagent | inx8_sgRNA1.R | this work | | AAACCCGATGCTACC AAGTCACTC Contact DG for more information |
| Recombinant DNA reagent (plasmid) | inx8_us_sgRNA1 | this work | | available upon request Contact DG for more information |
| sequence-based reagent | inx8_us_sgRNA1.F | this work | | TCTTGTGGAAAACAG AGGAATGGG Contact DG for more information |
| sequence-based reagent | inx8_us_sgRNA1.R | this work | | AAACCCCATTCCTCT GTTTTCCAC Contact DG for more information |
| sequence-based reagent | inx-14delF | this work | | GATACGACGTGAGCA ATGGAACGTC Contact DG for more information |

*Appendix 1 Continued on next page*

*Appendix 1 Continued*

| Reagent type (species) or resource | Designation | Source or reference | Identifiers | Additional information |
|---|---|---|---|---|
| sequence-based reagent | inx-14delR | this work | | CTTGGACTTGAAGT GAGAGTTGGAG Contact DG for more information |
| sequence-based reagent | sygl1-F | this work | | ATCATCGAACCA TTGTCATCACGC Contact DG for more information |
| sequence-based reagent | sygl1-R | this work | | TTTGCCTTGATCTC CAAGTGTTGC Contact DG for more information |
| Genetic reagent (*C. elegans*) | AG400 | *Starich et al., 2020* | fasn-1(av138[fasn-1::gfp]) I | |
| Genetic reagent (*C. elegans*) | CB190 | *Brenner, 1974* | unc-54(e190) I | |
| Genetic reagent (*C. elegans*) | CB1282 | *Hosono et al., 1982* | dpy-20(e1282) IV | |
| Genetic reagent (*C. elegans*) | DG2506 | *Govindan et al., 2009* | acy-4(ok1806) V; tnEx42[acy-4::gfp +rol-6(su1006)] | |
| Genetic reagent (*C. elegans*) | DG4959 | (this work) | qIs154[lag-2p::MYR::tdTomato +ttx-3p::gfp] V; tnIs5[lim-7p::gfp +rol-6(su1006)] X | Contact DG for more information |
| Genetic reagent (*C. elegans*) | DG4977 | (this work) | inx-14(ag17) I; qIs154[lag-2p::MYR::tdTomato +ttx-3p::gfp] V; tnIs5[lim-7p::gfp +rol-6(su1006)] X | Contact DG for more information |
| Genetic reagent (*C. elegans*) | DG5020 | (this work) | naIs37[lag-2p::mCherry::PH +unc-119(+)] I; bcIs39[lim-7p::ced-1::gfp +lin-15(+)] V | Contact DG for more information |
| Genetic reagent (*C. elegans*) | DG5026 | (this work) | inx-14(ag17) naIs37[lag-2p::mCherry::PH +unc-119(+)] I; bcIs39[lim-7p::ced-1::gfp +lin-15(+)] V | Contact DG for more information |
| Genetic reagent (*C. elegans*) | DG5027 | (this work) | naIs37[lag-2p::mCherry::PH +unc-119(+)] I; inx-9(ok1502) IV; bcIs39[lim-7p::ced-1::gfp +lin-15(+)] V | Contact DG for more information |
| Genetic reagent (*C. elegans*) | DG5029 | (this work) | naIs37[lag-2p::mCherry::PH +unc-119(+)] I; inx-8(tn1513tn1555) inx-9(ok1502) IV; bcIs39[lim-7p::ced-1::gfp +lin-15(+)] V | Contact DG for more information |
| Genetic reagent (*C. elegans*) | DG5059 | (this work) | inx-9(ok1502) IV | Contact DG for more information |
| Genetic reagent (*C. elegans*) | DG5063 | (this work) | inx-8(qy78[mKate2::inx-8]) IV | Contact DG for more information |
| Genetic reagent (*C. elegans*) | DG5064 | (this work) | inx-8(qy102(mKate2::inx-8)) inx-9(ok1502) IV | Contact DG for more information |
| Genetic reagent (*C. elegans*) | DG5070 | (this work) | inx-14(ag17) I; inx-8(qy78[mKate2::inx-8]) IV | Contact DG for more information |
| Genetic reagent (*C. elegans*) | DG5131 | (this work) | naIs37[lag-2p::mCherry::PH +unc-119(+)] I;inx-8(qy78[mKate2::inx-8]) IV; bcIs39[lim-7p::ced-1::gfp +lin-15(+)] V; | Contact DG for more information |
| Genetic reagent (*C. elegans*) | DG5133 | (this work) | naIs37[lag-2p::mCherry::PH +unc-119(+)] I; inx-8(qy102(mKate2::inx-8)) inx-9(ok1502) IV; bcIs39[lim-7p::ced-1::gfp +lin-15(+)] V | Contact DG for more information |
| Genetic reagent (*C. elegans*) | DG5136 | (this work) | sygl-1(q983[3xOLLAS::sygl-1]) naIs37[lag-2p::mCherry::PH +unc-119(+)] I; bcIs39[lim-7p::ced-1::gfp +lin-15(+)] V | Contact DG for more information |
| Genetic reagent (*C. elegans*) | DG5150 | (this work) | inx-14(ag17) sygl-1(q983[3xOLLAS::sygl-1]) naIs37[lag-2p::mCherry::PH +unc-119(+)] I; bcIs39[lim-7p::ced-1::gfp +lin-15(+)] V | Contact DG for more information |
| Genetic reagent (*C. elegans*) | DG5181 | (this work) | sygl-1(q983[3xOLLAS::sygl-1]) naIs37[lag-2p::mCherry::PH +unc-119(+)] I; inx-8(qy78[mKate2::inx-8]) IV; bcIs39[lim-7p::ced-1::gfp +lin-15(+)] V | Contact DG for more information |
| Genetic reagent (*C. elegans*) | DG5229 | (this work) | naIs37[lag-2p::mCherry::PH +unc-119(+)] I; inx-8(qy78tn2031) IV; bcIs39[lim-7p::ced-1::gfp +lin-15(+)] V | Contact DG for more information |

*Appendix 1 Continued on next page*

*Appendix 1 Continued*

| Reagent type (species) or resource | Designation | Source or reference | Identifiers | Additional information |
|---|---|---|---|---|
| Genetic reagent (*C. elegans*) | DG5232 | (this work) | naIs37[lag-2p::mCherry::PH +unc-119(+)] I; inx-8(tn2034) IV; bcIs39[lim-7p::ced-1::gfp +lin-15(+)] V | Contact DG for more information |
| Genetic reagent (*C. elegans*) | DG5248 | (this work) | sygl-1(q983[3xOLLAS::sygl-1]) naIs37[lag-2p::mCherry::PH +unc-119(+)] I; inx-8(qy78tn2031) IV; bcIs39[lim-7p::ced-1::gfp +lin-15(+)] V | Contact DG for more information |
| Genetic reagent (*C. elegans*) | DG5249 | (this work) | sygl-1(q983[3xOLLAS::sygl-1]) naIs37[lag-2p::mCherry::PH +unc-119(+)] I; inx-8(tn2034) IV; bcIs39[lim-7p::ced-1::gfp +lin-15(+)] V | Contact DG for more information |
| Genetic reagent (*C. elegans*) | DG5250 | (this work) | inx-8(qy78tn2031) IV | Contact DG for more information |
| Genetic reagent (*C. elegans*) | DG5251 | (this work) | inx-8(tn2034) IV | Contact DG for more information |
| Genetic reagent (*C. elegans*) | DG5270 | (this work) | inx-14(ag17) I | Contact DG for more information |
| Genetic reagent (*C. elegans*) | DG5310 | (this work) | naIs37[lag-2p::mCherry::PH +unc-119(+)] I; acy-4(ok1806) V; tnEx42[acy-4::gfp +rol-6(su1006)] | Contact DG for more information |
| Genetic reagent (*C. elegans*) | DG5320 | (this work) | fasn-1(av138[fasn-1::gfp]) naIs37[lag-2p::mCherry::PH +unc-119(+)] I | Contact DG for more information |
| Genetic reagent (*C. elegans*) | DG5346 | (this work) | naIs37[lag-2p::mCherry::PH +unc-119(+)] I; inx-8(qy78[mKate2::inx-8])/tmC5[tmIs1220] IV; bcIs39[lim-7p::ced-1::gfp +lin-15(+)] V | Contact DG for more information |
| Genetic reagent (*C. elegans*) | DG5347 | (this work) | naIs37[lag-2p::mCherry::PH +unc-119(+)] I; inx-8(qy78tn2031)/tmC5[tmIs1220] IV; bcIs39[lim-7p::ced-1::gfp +lin-15(+)] V | Contact DG for more information |
| Genetic reagent (*C. elegans*) | DG5357 | (this work) | tmC5[tmIs1220] inx-8(tn2075) IV | Contact DG for more information |
| Genetic reagent (*C. elegans*) | DG5366 | (this work) | naIs37[lag-2p::mCherry::PH +unc-119(+)] I; inx-8(qy78[mKate2::inx-8])/tmC5[tmIs1220] inx-8(tn2075) IV; bcIs39[lim-7p::ced-1::gfp +lin-15(+)] V | Contact DG for more information |
| Genetic reagent (*C. elegans*) | DG5367 | (this work) | inx-14(ag17) fasn-1(av138[fasn-1::gfp]) naIs37[lag-2p::mCherry::PH +unc-119(+)] I | Contact DG for more information |
| Genetic reagent (*C. elegans*) | DG5378 | (this work) | fasn-1(av138[fasn-1::gfp]) naIs37[lag-2p::mCherry::PH +unc-119(+)] I; inx-8(qy78[mKate2::inx-8]) IV | Contact DG for more information |
| Genetic reagent (*C. elegans*) | DG5380 | (this work) | bcIs39[lim-7p::ced-1::gfp +lin-15(+)] V | Contact DG for more information |
| Genetic reagent (*C. elegans*) | FX30140 | *Dejima et al., 2018* | tmC5[tmIs1220] IV | Contact DG for more information |
| Genetic reagent (*C. elegans*) | JK1466 | *Francis et al., 1995* | gld-1(q485)/dpy-5(e61) unc-32(e51) I | Contact DG for more information |
| Genetic reagent (*C. elegans*) | KLG006 | *Gordon et al., 2020* | inx-8(qy78[mKate2::inx-8]) IV; tnIs6[plim-7::gfp +rol-6(su1006)] X; cpIs122(lag-2p::mNeonGreen::plcdeltaPH) | Contact DG for more information |
| Genetic reagent (*C. elegans*) | NK2571 | *Gordon et al., 2020* | inx-8(qy78[mKate2::inx-8]); cpIs122 [lag-2p::mNeonGreen::plcdeltaPH] | Contact DG for more information |
| Genetic reagent (*C. elegans*) | NK2576 | *Gordon et al., 2020* | inx-8(qy102(mKate2::inx-8)) inx-9(ok1502); cpIs122(lag-2p::mNeonGreen::plcdeltaPH) | Contact DG for more information |

