## [Editor Report]

This manuscript is important and of interest to a broad audience of stem cell scientists interested in the regulation of stem cell niche architecture. The authors provide a detailed description of the effects of the loss of innexin, a gap junction protein, on the somatic cells that support the distal, proliferative end of the *C. elegans* germ line. The key findings of this manuscript pose a strong challenge to a recent new model of somatic sheath Sh1 cells determining the boundary of germline stem cell niche compartment. Discrepancies remain between the current manuscript and the manuscript by Li et al., which must be resolved in the future.

---

## [Decision Letter]

**Decision letter after peer review:**

Thank you for submitting your article "Innexin function dictates the spatial relationship between distal somatic cells in the *Caenorhabditis elegans* gonad without impacting the germline stem cell pool" for consideration by *eLife*. We apologize for the delay in reaching the decision.

Your article has been reviewed by 3 peer reviewers, including Yukiko M Yamashita as the Reviewing Editor and Reviewer #3, and the evaluation has been overseen by a Reviewing Editor and Anna Akhmanova as the Senior Editor. The following individuals involved in review of your submission have agreed to reveal their identity: Ekaterina Voronina (Reviewer #1); Judith Yanowitz (Reviewer #2).

Essential revisions:

Overall, the paper was reviewed favorably, and the paper is likely worthy of the publication in *eLife*. One major comment that requires additional experiment(s) is the clarification of inx-8-mkate allele. The interpretation of poisonous/antimorph/hypomorph etc is influenced by whether or not the allele was homozygote or heterozygote, but such information is not clearly provided in the manuscript that allows appropriate interpretation.

Furthermore, while the reviewers were discussing the decision of your manuscript, the other paper was submitted to *eLife* by Kacy Gordon's group, which is already posted on BioRxiv. https://www.biorxiv.org/content/10.1101/2021.11.08.467787v2

You will see that they reached a different conclusion. We reviewers agreed that some discrepancies may have derived from different markers, imaging conditions etc. Each paper will be reviewed for its own merit fairly, but as a journal, we need to ask both parties not to leave simple discrepancies that are caused by easy-to-address differences such as reagents and imaging conditions. Also, given that the different makers may be giving different results, we ask you to label the figure panels correctly (e.g. state the markers used, instead of 'what it is supposed to show (such as Sh1)') We ask both parties to communicate and put best effort into resolving discrepancies.

Other comments provided by reviewers are listed below, which we believe can be addressed mostly by textual editing.

*Reviewer #1 (Recommendations for the authors):*

The authors used mean DTC process length to estimate DTC location, which allowed quantitative analysis of cellular organization, but does not directly challenge the "interdigitated" morphology proposed by Gordon et al. However, investigation of several innexin reduction of function mutants provides compelling evidence that the relative DTC/Sh1 positions are impacted by gap junctions between the germ and somatic cells. Furthermore, the conclusions that the changes in relative DTC/Sh1 positions do not concordantly affect the germ line stem cell pool size are strong and bring critical insight to the field.

*Reviewer #2 (Recommendations for the authors):*

There are several suggestions below for experiments or experimental details that would improve the manuscript.

1) The authors should clarify the relationship between the DTC and SH1 and whether innexin function is needed in one or both. The authors now show that there is overlap in the DTC and Sh1 marked regions. Are these cells growing over one another or are the projections intercalated? Can this be shown by EM or super-resolution imaging? Does the loss of innexin function in just Sh1 or the DTC change the border? This might help resolve their model about whether Sh1 is "sensing" reduced gap function number or function?

2) I would be reluctant to claim that they have "Consistent trends… from multiple markers." (line 194) Ideally, they would use different promoters expressed in the same tissue. If this cannot be done because other well characterized promoters do not exist, then the authors need to say, "Consistent results are obtained from independent transgenes"? Similarly in the paragraph line 219 -226, it is unclear what you meant to test here is the strain was first tested with one of the transgenes.

3) The authors argue that the qy78 fusion protein acts an antimorph and a loss-of-function to rigorously test these statements they should show the behaviour of qy78/+ and qy78/null. This would allow them to determine if it is a hypermorph, antimorph, or neomorph and if truly recessive.

4) qy78tn2031 appears to be slightly more abnormal than the tn2034 (Figure 2-Suppl 2B) It might be worth acknowledging that the slight difference could be due to strain background variation? Or are there other reasons for this difference?

5) Lines 252-254: From table 1, it does not appear that inx-14 suppresses qy78. Also, the interpretation that inx-9 suppresses is also complicated by the fact that the original qy78 alleles gave brood sizes of ~155 whereas the outcross to the DG strains have lower brood sizes. This needs to be clarified.

6) Line 326: Re "no significant change in the size of the SYGL-1-positive stem cell pool". The authors are not quantifying either the size of the stained region or (more appropriately) the number of nuclei that have SYGL-1 staining, just the extent of reach of the marker. If anything, a subset of the mutant germ lines (Figure 3) appear smaller than control. This should be addressed.

7) Similarly in this entire section, while it is clear that Sh1 is shifted distally (almost to the distal end of the germ line), it would be nice to know how the authors quantify proximal shift since the data is in microns which are not normalized for the total size of the germ line. Is there a change in the total number of nuclei or the numbers of rows?

8) No mention is made in the text of Figure 4 qy78tn2031 or tn2034 or of Figure 4D or Figure 4 figure supplement 1

9) Table 1 is very confusing because so much of the information in the footnotes refers to methods. It is also unclear, but does b represent the cumulative data from 2 different outcrosses both of which decreased brood size? Do they think there is a suppressor in the original strain? Either all the info in Table 1 footnotes should be included in a supplemental table or in the methods.

Also, it might be helpful to consolidate a lot of the information into one large table-specifically added a column to Table 1 that includes a yes/no or distance for positional information of Sh1. (maybe "Sh1 edge moved distally")

10) The graphics of the genes and mutations should be combined into one figure.

There were a number of textual edit that need to be made:

Textual Edits

Line 241: "must not appreciably perturb the function of inx-9" Please clarify why you think this might be the case?

Lines 833-834: This parenthetical about pSUN-1 pSer8 not being shown should be included in the figure legends since it is mentioned in the text as being used to define the border and is not shown in the figure. Or simply add "(see methods)" after the "pSUN-1(-) progenitor pool".

Line 52: "Germ line" should be one word.

Line 57: need a comma after "(DTC)".

Line 62: Additional support for what?

Line 57 and 63: Figure 1C (just listed as Figure 1) is mentioned before Figures 1A and 1B.

Figure 1A: white "inx-21" hard to read on the yellow background. Also, the proteins names should be capitalized not lower case.

Line 80: "germline" should be "germ line".

Line 81-83: The lof analysis does not follow from the rescue study. These should be separate sentences.

Line 92: Should read "Figure 1C and D".

Lines 155-156: the order of inx-8 and inx-9 needs to be reversed (written 9 then 8).

Line 169: remove comma after: "2020))".

Line 170-171: either remove comma before "as well as" or add comma after "inx-8 mutants".

Line 216: The parenthetical is confusing as it comes after reference to inx-9 null allele, but refers to inx-8.

Lines 223-225: the sentences are repeated.

Line 226: do you have a reference for this (although common knowledge in the worm field it would be helpful). This is however integrated at the inx-8 locus, so it would be helpful to comment on and reference why this might be seen with single copy integrants.

Line 244: Rather that "further" perhaps say "in addition" or "moreover". Also, the space between the alleles "qy78 tn2031" should be deleted. (also in Figure 2A).

Line 250: I would say "adhesion functions" rather than "rivet functions" since the latter is an analogy to describe the adhesion role. Also "germline" should be "germ line" here and throughout since soma is a noun, presumably germ line should be as well.

Lines 323-324: reference for the sygl-1(OLLAS) should be provided.

Line 326: For consistence you should write "inx-8(qy78[mKate2::inx-8])" throughout or "inx-8(qy78)" but not both. Personally, the former is more helpful is reading the text.

Line 333: "each other" should be "one another".

Line 343: "1-2 cell diameters". What is the basis of this claim?

Line 439: Add a comma after "organ system".

Line 495: is there a funding stream for Wormbase that can be cited?

Lines 686-687: I am presuming the chromosome insertion sites for cpIs122 and naIs37 are not known?

Line 832: "until" Not "till".

*Reviewer#3 (Recommendations for the authors):*

Explanation of Figure 2 is a bit confusing. When they used CRISPR approach to remove poisonous inx-8-mKate gene, is the animal examined homozygous for inx-8 deletion, or heterozygous over wild type allele? If the claim of inx-8-mKate being poisonous is correct, inx-8-mKate/+ would also have phenotypes (reduced brood size, loss of the bare region), and the removal of mKate allele (in the presence of inx-8 wild type allele) should rescue the phenotype entirely. Alternatively, if the authors mean that inx-8-mkate's phenotype can be only observed as homozygote, it might mean that inx-8-mKate is a loss of function allele. Also, I see the possibility that the authors imply that inx8-mKate is poisonous to inx-9, and poisoning effect can be seen only as homozygote? Whereas the data itself is convincing, the terminology (poisonous = dominant-negative, hypomorph) as well as the description of genetics (heterozygote vs. homozygote) that can guide the determination of poisonous vs. hypomorph, is unclear to the readers. This should be clarified.

[Editors' note: further revisions were suggested prior to acceptance, as described below.]

Thank you for resubmitting your work entitled "Innexin function dictates the spatial relationship between distal somatic cells in the *Caenorhabditis elegans* gonad without impacting the germline stem cell pool" for further consideration by *eLife*. Your revised article has been evaluated by Anna Akhmanova (Senior Editor) and a Reviewing Editor.

The manuscript has been improved but there are some remaining issues that need to be addressed, as you can see individual reviewer comments (which are all minor, and can be addressed by textual changes).

*Reviewer #1 (Recommendations for the authors):*

In the revised version of the manuscript, Tolkin et al. addressed all of reviewers' suggestions and included new data that significantly strengthened the manuscript. Especially notable are: 1) the inclusion of the new classification of DTC/Sh1 process positioning. It is very helpful to address morphological variation and to allow clear comparison among different strains and markers of DTC and Sh1. 2) the inclusion of immunostaining with anti-INX-8. This clearly removes the criticism of sub-optimal markers or a concern that distinct Sh1 boundary locations are attributable to using different markers. Anti-INX-8 clearly shows distinct locations of Sh1 boundary in the wild type vs inx-8(qy78[mKate::inx-8]) backgrounds.

Overall, this work challenges the Gordon et al., 2020 model of oriented cell divisions in *C. elegans* germline and presents a significant advance in the field.

*Reviewer #2 (Recommendations for the authors):*

The revised manuscript addresses all of the major concerns from the previous submission by using multiple reporters for each cell type, by creating a classification system, by providing EM reconstruction of the DTC/SH1 cell and processes, and by redoing all of the viability assays to address strain differences between labs. The importance of the paper remains both in its tale of woe for transgenic reporters, but more importantly, by providing a framework for detailed characterization and classification of cell:cell interactions. It also raises new and interesting questions about the roles of the sheath cells and that innexin proteins might slight haploinsufficiency, a phenotype that may be relevant to human disease.

There are minor changes to the text that should be addressed

Line 138- 139. Technically the images in Figures 1 and suppl do not show the germ cells proper. Move the comment about DIC imaging from Line 163 to line 138 for clarification.

Lines 261-267: The change in the allele names here is confusing and would help it were described in the text itself rather than just the Materials and methods.

There are two odd source file to PDF conversion issues that you should check carefully is accepted for publication:

• In all figures in the PDF, the gene names and alleles are not italicized in all figures and graphs. But in the source files, they seemed correct (italic)!

• In Figure 1B and 2A, the text of the alleles appears to be shadowed oddly (in the PDF- but not the source files). Please confirm there is not an issue with the text in that figure. Also in Figure 1 Suppl 1A, the pink text.

p. 3 line 1 add comma after interactions.

Line 83-84 need a reference. Is this the paper you are refuting or other? It seems you are saying that Sh1 drives cell fate determining germ cell divisions, since you said that ectopic inx-8 in the Sh1 in inx8(0) inx-9(0) rescued proliferation. Isn't this contradicting what you are setting out to prove?

Line 138 appears to be missing words or punctuation.

Line 168, "wild-type unmarked young adult" add come after "wild-type".

Kerning of text in Figure 2 Suppl 1 and Suppl 2B above the Figuresis off (again in my PDF, but is this true for all?)

Line 285 "…in one but not both cells.." should read "in one, but not both, cells".

Line 320, the statement "that are lacking in inx-8(qy78[mKate2::inx-8])" is confusing and makes it seem that you are saying it is lacking in mKate (which it is due to the deletion) not that it dissimilar from qy78 which I what I think you intended. I would just remove this part of the sentence.

Line 381 need reference to a figure.

Line 385, italicize "per se".

Line 499, italicize "*Xenopus*".

Line 819 "Embryonic" should be lower case.

There should be no hyphen for "P-value". And the "P" should be italicized "P".

*Reviewer #3 (Recommendations for the authors):*

This manuscript by Tolkin et al. provide the evidence that a previously-published *eLife* paper by Gordon et al. was misled by the use of inx-8-GFP, which Tolkin found to function as a dominant negative form. Accordingly, this manuscript substantially changes our current understanding of how germline stem cells and their differentiation are regulated in the *C. elegans* gonad.

Whereas there remain some unresolved discrepancies between this manuscript and Gordon et al., the revised version of Tolkin et al. manuscript strongly supports their main conclusions, including the presence of 'bare region' (the area of germ cells that are not touched by Sheath cells or Distal Tip Cells), and the germ cell biology supported by such an architecture of the gonad. Even if some discrepancies may end up to be corrected in the future, their main conclusion is firm enough that merits publication in *eLife*.

Whereas their conclusion 'Sh1 position is neither relevant for GLP-1/Notch signaling nor for the exit of germ cells from the stem cell pool' (in the abstract) is correct in that the junction of DTC and Sh1 cells do not serve as the interface for GSCs' 'asymmetric division' (as Gordon 2020 paper suggested), it seems that inx influences the position of Sh1 and also impacts brood size (maybe they think 250 vs. 210 are not different, but have they run statistics on these numbers? It looks to me that there are correlation between Sh1 position and a slight reduction in brood size?). If there is any impact of inx8 mutant/dominant negative, it should be a bit more carefully described. I recommend revising the text to more accurately describe their important conclusions (that innexin mutant reduces bare region with likely functional consequences).

---

## [Author Response]

Essential revisions:Overall, the paper was reviewed favorably, and the paper is likely worthy of the publication in eLife. One major comment that requires additional experiment(s) is the clarification of inx-8-mkate allele. The interpretation of poisonous/antimorph/hypomorph etc is influenced by whether or not the allele was homozygote or heterozygote, but such information is not clearly provided in the manuscript that allows appropriate interpretation.

See below, we built the necessary strains and addressed this point. The dosage studies support the interpretation that the *inx-8(qy78[mKate2::inx-8])* is a dominant antimorph that poisons innexin function.

Furthermore, while the reviewers were discussing the decision of your manuscript, the other paper was submitted to eLife by Kacy Gordon's group, which is already posted on BioRxiv. https://www.biorxiv.org/content/10.1101/2021.11.08.467787v2You will see that they reached a different conclusion. We reviewers agreed that some discrepancies may have derived from different markers, imaging conditions etc. Each paper will be reviewed for its own merit fairly, but as a journal, we need to ask both parties not to leave simple discrepancies that are caused by easy-to-address differences such as reagents and imaging conditions.

We see these apparent discrepancies as falling into two areas:

(1) brood size and embryonic lethality and

(2) “bare regions” in the wild type.

(1) brood size and embryonic lethality

For strains bearing the *inx-8(qy78[mKate2::inx-8])* allele, we have partially reconciled with the Gordon lab results reported in the Li et al., 2021 pre-print.

For reasons we have yet to fully resolve, in our hands *inx-8(qy78[mKate2::inx-8])* brood sizes and embryonic lethality can be quite variable. The most dramatic reductions in brood size and increases in embryonic lethality may be linked to recovery from stress; however, we have not verified this as the cause. Parenthetically, we note that *inx-8(tn1513tn1555)*, a reduction-of-function allele (T. Starich, unpublished results), recovers fertility poorly after prolonged starvation. The penetrance of this sterility is also variable (50–90%). Interestingly, and perhaps relevant to our observations with the INX-8 fusion, the Li et al. preprint reports that the KLG019 strain, which contains an N-terminal fusion of GFP with INX-9*,* exhibits a high level of embryonic lethality.

Once we fully appreciated this unusual situation, we repeated all of the brood and embryo counts with very strictly controlled conditions and report here in revised Table 1. We mention this in the context of other observations made from our brood size and embryonic viability analysis: “Third, in the course of this analysis, we noted that strains bearing *inx-8(qy78[mKate2::inx-8]* displayed highly variable population growth dynamics that we attribute to stress. Although the relevant stressor has yet to be determined, we observed this effect over multiple generations after starvation or transport. Given our previous finding that key metabolites such as malonyl-CoA are transported through these junctions to support embryonic growth, this finding deserves future scrutiny.”

In short, for *inx-8(qy78[mKate2::inx-8],* our results are more similar to those of Li et al. preprint, and these results correct our *bioRxiv* report of high embryonic lethality. We considered putting Table 1 reporting the brood/viability information into the supplement but opted to keep it in the main text to most visibly correct the error in our *bioRxiv* report.

Nevertheless, *inx-8(qy78[mKate2::inx-8]* does not behave like the wild type. The brood size is lower than the wild type, and there is some embryonic lethality (likely an effect of altered function of proximal sheath–oocyte gap junctions). Our analysis also shows that embryonic lethality is enhanced in combination with a hypomorphic allele of *inx-14* [*inx-14(ag17)*] and appears suppressed in combination with a null mutation in *inx-9* [*inx-9(ok1502)*]. This behavior is consistent with the idea that the mKate2::INX-8 fusion alters the function of the channels in which it is incorporated since *inx-8* intragenic null mutations ameliorate these effects.

We are still surprised by the low brood and high embryonic lethality defects reported by Li et al. for strains bearing *bcIs39*. We do not observe these phenotypes, neither in some of the same strains they examined, nor in a strain with *bcIs39* alone. We offered to resend/send all strains to Dr. Gordon.

The more important point is that in properly-controlled experiments (that is, where innexin mutants are being compared to strains bearing wild-type innexin genes and the same markers for each), we see similar effects on the distal border of Sh1 regardless of the Sh1 markers used.

Brood sizes and viability phenotypes remain secondary to the question of Sh1 position and its influence on germline stem cells.

(2) “bare regions” in the wild type

This, unfortunately, is not a “simple discrepancy.”

We stand by our conclusion that the loss of the bare regions, which they (and we) observe in strains carrying mKate2::INX-8, is a direct result of the distal mispositioning of Sh1 caused by altered innexin function. We pointed out this possibility to Dr. Gordon on several occasions prior to our initial submission.

By assuming that the Sh1 position in the *inx-8(qy78[mKate2::inx-8])* strain was the “normal” position (counter to previously published results), mKate2 expression was interpreted as visualizing new Sh1 membrane not seen using previous markers. When analyzing live mKate2::INX-8-bearing strains, we also see that the minimum distance between DTC and Sh1 is often about 1–2 cell diameters. This distance could give the impression that Sh1 might direct asymmetric division of GSCs. Nevertheless, without observing the positional relationship between Sh1 and the direct GLP-1/Notch target SYGL-1, the claim that Sh1 directs asymmetric division remained weak.

As summarized below, we have further characterized the natural variability in Sh1 position in the wild type, introducing a phenotypic classification scheme to compare genotypes (2.1), and we bolstered our conclusions from the original manuscript with additional strains and independent techniques (2.2).

(2.1) In our revised manuscript, we introduce a phenotypic classification scheme (see revised Figure 1, Figure 1—figure supplement 1, Figure 2, Figure 2—figure supplement 2, Figure 2—figure supplement 3) that helps convey:

(i) the natural variability in the position of Sh1 in the wild type, as visualized with different markers,

(ii) that Sh1 is often many germ cell diameters distance away from the DTC (an important point that argues against the Gordon (2020) model of oriented cell divisions),

(iii) that several different alterations in innexin function result in similar distal mis-positioning of Sh1, and

(iv) that *inx-8(qy78[mKate2::INX-8]* produces a dominantly acting toxic product.

(2.2) In our revised manuscript, as detailed in sections (2.2.1-3) below, we use additional live markers (including one generated using CRISPR-Cas9 genome editing, to argue against the concern of multicopy transgenes), we perform immunohistochemistry in the wild type (no markers present), and present analysis of serial sectioning transmission electron microscopy using high-pressure freezing and freeze-substitution, to study the position and arrangement of Sh1, the DTC, and germ cells in the distal gonad of the wild type and innexin mutants.

We were disappointed that the Li et al. preprint did not address our observation that Sh1 is almost invariantly mispositioned (distally) in *inx-8(qy78[mKate2::inx-8]),* the strain used as a “marker” for Sh1, and that this is a phenotype shared by another *inx-8* mutant and by *inx-14(ag17)*. Instead, their preprint focused on variability in Sh1 position between markers, and suggested that the markers we used failed to illuminate the distal part of Sh1. It is well known in the field (as they note) that marker strains can be variable—both in expression level and in their effects on specific phenotypes. That is why one compares genotypes only between similarly marked strains.

However, there is an unfortunate logical flaw in their argument. If it were the case that our markers fail to detect the distal part of Sh1, we should not have been able to quantify a significant distal shift of the Sh1 boundary in several different live markers in innexin-deficient backgrounds that we document in this manuscript (Figure 1, Figure 2, Figure 2—figure supplement 2, Figure 2—figure supplement 3).

All that said, no markers are perfect. As we showed in our original submission, the frequency of perfect overlap between *inx-8(qy78[mKate::inx-8])* and *bcIs39(lim-7p::CED-1::GFP)* or between *inx-8(qy78[mKate::inx-8])* and *tnIs6(lim-7p::GFP)* is 85% (not 100%). Nevertheless, the frequency of distal placement of Sh1 increases significantly in the *inx-8(qy78[mKate2::inx-8)])* background. In revised Figure 2—figure supplement 3D, we provide results obtained by scoring images using mKate2 *or* using GFP, in strains carrying both *inx-8(qy78[mKate2::inx-8])* and concurrently-expressed *lim-7p*-based GFP markers. While the GFP strains modestly underestimate Sh1 mis-positioning, the penetrance of mis-positioning is nevertheless striking.

(2.2.1) In our revision, we add analysis of live worms bearing two non-*lim-7* promoter-driven markers. These display the expected wild-type Sh1 position and have normal broods, (or even more progeny, as in the case of the single-copy CRISPR-Cas9 generated *fasn-1(av138[GFP])*. In *inx-14(ag17)* we see a distally mispositioned Sh1 but no viability defects. These points are added to Figure 1—figure supplement 1 and the revised manuscript text.

(2.2.2) In our revision, we add an anti-INX-8 antibody staining analysis in the wild type with no markers at all and in the *inx-8(qy78)* allele (revised Figure 2B and C; Figure 2—figure supplement 1). This analysis shows that the distal boundary of INX-8 expression is significantly closer to the distal end of the gonad in the *inx-8(qy78[mKate2::inx-8)])* (homozygous) genetic background than in a wild-type background with no additional markers.

In sum, regardless of whether markers do or do not detect the distal part of Sh1, the anti-INX-8 antibody staining should have shown no significant difference in the distal boundary of Sh1 between *inx-8(qy78[mKate2::inx-8])* and *inx-8(+)* if mKate2::INX-8 is marking the wild-type boundary of Sh1.

(2.2.3) In our revision we add EM analysis that demonstrates that a wild-type distal gonad contains bare regions in which germ cells do not contact either the DTC or Sh1 and are covered only by a basal lamina, as described in Hall et al., (1999). We have added David Hall and Ken Nguyen to the author list as they contributed the EM analysis. Our study is consistent with and extends our previously published work and corrects a misconception in the field that has recently taken root because of the 2020 *eLife* report.

Regarding the new EM analysis: it would have been difficult to do a similar analysis in 1999 because all images had to be taken on glass negatives. Even with a digital camera today, this was laborious and technically demanding work. However, we do now provide a more extensive description of the variability in the length and arrangement of the bare regions using confocal microscopy and image reconstructions. Taken together, these findings argue against the model that interactions with Sh1 promote stem cell niche exit and differentiation of germ cells as proposed by Gordon et al., (2020).

In our revision, we also present additional new findings showing that *inx-8(qy78[mKate2::inx-8)])* behaves as a dominant antimorphic mutant that interferes with germline–soma gap junctional activity (Figure 2—figure supplement 2).

Finally, and of importance to the central question of all of these manuscripts, if Sh1 played a key role in directing differentiation of germline stem cells, we should have observed concomitant changes in the border of SYGL-1 expression between innexin mutants and the wild type, which we do not. Neither Gordon et al. (2020) nor Li et al., (2021) *bioRxiv* pre-print examined the position of the stem cell border relative to Sh1.

Also, given that the different makers may be giving different results, we ask you to label the figure panels correctly (e.g. state the markers used, instead of 'what it is supposed to show (such as Sh1)').

There is no disagreement between us or with any others in the field, including Dr. Gordon, that the markers mark the cells of interest (DTC and Sh1). The only “disagreement,” as noted above, is whether the Sh1 markers we use marks “all” of this very large Sh1 pair of cells. Therefore, we have opted to indicate both the marker names and “DTC” and “Sh1” on each figure as this helps the reader to follow the results as described in the text, especially as we present data using 2 different DTC markers and 4 different Sh1 markers. In sum, all figures clearly indicate the marker used.

We ask both parties to communicate and put best effort into resolving discrepancies.

Prior to our original submission, we communicated extensively with Dr. Gordon. We obtained the *inx-8(qy78[mKate2::inx-8]*-bearing strain directly from her. It is also worth pointing out that effect of *inx-14(ag17)* on the distal extent of Sh1 was noted by Starich et al., (2014, see Figure 7D). Since the editors have emphasized the importance of communication between our groups, we would like to alert the editors that in an email from June 14, 2020 one of us (DG) alerted Dr. Gordon that loss of innexin function in the germline or sheath cells could cause Sh1 to extend to the distal tip, based on the aforementioned published studies. We similarly communicated with her as we obtained results presented herein, and, as noted in the Li et al. preprint, we sent all our strains as well as our original submission.

Finally, we have posted our revised manuscript on *bioRxiv* and notified Dr. Gordon. We have offered to send all strains, and have offered to meet and discuss our findings.

Other comments provided by reviewers are listed below, which we believe can be addressed mostly by textual editing.Reviewer #1 (Recommendations for the authors):The authors used mean DTC process length to estimate DTC location, which allowed quantitative analysis of cellular organization, but does not directly challenge the "interdigitated" morphology proposed by Gordon et al. However, investigation of several innexin reduction of function mutants provides compelling evidence that the relative DTC/Sh1 positions are impacted by gap junctions between the germ and somatic cells. Furthermore, the conclusions that the changes in relative DTC/Sh1 positions do not concordantly affect the germ line stem cell pool size are strong and bring critical insight to the field.

We thank the reviewer for the positive comments.

Our findings support our previous claim that there are “bare regions” between the proximal end of the DTC and distal end of Sh1 (Hall et al., 1999). In the 1999 paper, we noted that the distal filopodia of Sh1 reach approximately 19 ± 6 germ cell diameters from the distal tip of the gonad (p. 113), findings that are essentially replicated here. However, in that 1999 paper, we did note that “the precise length and arrangement of the bare regions seem to be variable (legend to Figure 11B, p. 117).” In 1999, we lacked the tools to describe in a more comprehensive manner the variability in the bare regions because we were unable to mark the DTC and Sh1 simultaneously with different colored fluorophores.

A key contribution of our revised manuscript is that we have now described the variability of the bare regions through extensive confocal analyses. These results are also supported by a new EM reconstruction using high-pressure freezing and freeze-substitution preparation methods. Yet, this natural variability is perhaps most easily seen plotted in revised Figure 1G and subsequent “barbell” plots of that kind and in the green dots representing the position of the distal border of Sh1 as visualized with *bcIs39*, *tnIs5*, *fasn-1(av138[GFP])* and *tnEx42* (revised Figure 1 and Figure 1—figure supplement 1). Furthermore, we see a highly variable Sh1 position as marked by anti-INX-8 antibody in the wild-type without any markers (revised Figure 2B and C), often positioned at over 100 µm from the distal end, far proximal to the position occupied by DTC processes (as seen with DTC-marked strains in Figure 1—figure supplement 1 panel E).

Supporting the results presented in the first submission and addressing reviewers’ concerns, after extensive analysis (including a new slice-by-slice analysis of our live images, collection of additional live images, EM reconstruction, additional antibody staining, and analysis of additional markers) we settled on a new classification scheme to better define and quantify the variability we see. In short, we stratified our images into 3 classes based on the position of the distal-most border of Sh1 (indicated in all figures by a green dot) relative to the proximal reach of the DTC (indicate in all figures by a pink dot), the mean reach of DTC processes (indicated by a blue dot – now also added to all “barbell” graphs for each gonad), and the distal end of the gonad.

This new classification scheme highlights a key finding of our work: that alterations of *inx-14* or *inx-8* activity dramatically shift the border of Sh1 toward the distal end of the gonad.

We defined phenotypic classes as follows based on the position of the DTC and Sh1

(see revised Figure 1C-F):

As shown in revised Figure 1F, when we classified wild-type gonads (that is, *inx-x(+)* with markers), ~70% are in Class 1, whereas the vast majority of gonads in strains bearing reduction of *inx-14* or *inx-8* function are in Class 3, regardless of marker combinations (revised Figure 1 —figure supplement 1 and Figure 2). Measurements are broken down by individual gonad (e.g., revised Figure 1 panel G, as in the original Figure 1), but with the addition of a blue dot on each gonad representing the position of the mean length of DTC processes.

In new Videos 1–3, we provide 3-dimensional renderings (made with Imaris) of gonad arms representing each phenotypic class.

In a new Figure 1—figure supplement 2, we provide an EM reconstruction of a gonad arm from a young adult wild-type hermaphrodite (unmarked strain), sliced from distal to proximal, outlining the DTC and Sh1. The images and reconstruction are provided in new videos 4 and 5, with highlights of germ cells that are/are not in contact with the DTC or with Sh1.

In revised Figure 1—figure supplement 1, we compare the results shown in Figure 1 with alternate DTC and Sh1 markers, including *tnIs5[lim-7p::GFP]* and newly constructed and analyzed strains bearing other markers, (including two that are not driven by *lim-7p*) and/or *inx-14(ag17)*.

While different markers do alter the *exact* proportion of Class 1 (gap) and Class 2 (small gap/intercalation) gonad arms, *regardless of the markers used*, reducing the activity of *inx-14 dramatically* shifts the distal border of Sh1 toward the distal end of the gonad such that almost 90–100% of the gonads are Class 3 (distal Sh1 position) (revised Figure 1— figure supplement 2 panel C). Measurements are broken down by individual gonad where the position of each Sh1 border can be seen in panel D and averages with statistics in panel E.

In revised Figure 2D–F, we apply the same classification scheme to *inx-8(qy78[mKate2::inx-8])*, and to the strain in which the *inx-8* fusion has been deleted. Here, we see that, measuring with *bcIs39* [*lim-7p::ced-1::GFP*], the strains bearing *inx-8(qy78[mKate2::inx-8])* display a reproducible shift of the Sh1 border to the distal end. Moreover, when the fusion-protein (mKate2::INX-8) coding sequence is removed in *inx-8(qy78tn2031)*, the distal Sh1 shift is suppressed.

In revised Figure 2B–C, we present new information further supporting this conclusion. In *inx-8(qy78[mKate2::inx-8])*, with no other markers present, as detected by immunohistochemistry with anti-INX-8 antibodies, the distal border of Sh1 is shifted distally relative to the wild type.

Supporting our description of the nature of *inx-8(qy78[mKate2::inx-8]* as a “poisonous” allele, new Figure 2—figure supplement 2 provides a similar analysis as in Figures 1 and 2 for additional strains, including *inx-8(qy78)* (or the equivalent *qy102*), with and without *inx-9 mutations*, as well as *inx-8(qy78)/+* and *inx-8(qy78)/inx-8(0).* In short, these data show that *qy78* is a dominant allele that, because of its similar phenotype to reduction of innexin function scenarios (e.g., *inx-14(ag17)* and *“inx-8(rf)”*), we could consider an antimorphic behavior.

However, it is not simple. Although the presence or absence of *inx-9* (that is partially redundant with *inx-8*) does not affect the phenotype of *inx-8(qy102)* (which is equivalent to *qy78*), the phenotype of *inx-8(qy78)/+* is far more severe than *inx-8(qy78)/inx-8(0)*, indicating that the dosage of the mutant allele relative to wild-type is important. In the revised manuscript, we comment on this point and its likely relevance to the multimeric structure of the innexins.

Revised Figure 2—figure supplement 3 panels A–C retain the overlap analysis for *inx-8(qy78[mKate2::inx-8]* and *bcIs39* [*lim-7p::CED-1::GFP*]. We added a new panel D with the slice-by-slice classification analysis performed using either *inx-8(qy78[mKate2::inx-8]* and *bcIs39* [*lim-7p::CED-1::GFP*] or *inx-8(qy78[mKate2::inx-8]* and *tnIs6(lim-7p::GFP)* as the Sh1 marker.

The results, once again, demonstrate that *inx-8(qy78[mKate2::inx-8]* virtually eliminates Class 1 and shifts the distally the border of Sh1 the majority of the time.

Reviewer #2 (Recommendations for the authors):There are several suggestions below for experiments or experimental details that would improve the manuscript.1) The authors should clarify the relationship between the DTC and SH1 and whether innexin function is needed in one or both.

Figure 12A of Starich et al., (2014) shows an *inx-8(0) inx-9(0)* double null mutant in which *inx-8::gfp* is expressed in the DTC using the *lag-2* promoter. This expression substantially restores germline proliferation to the *inx-8(0) inx-9(0)* double null mutant. In such “mosaic animals,” sheath cells are unable to form gap junctions with germ cells, but the DTC can. As seen in the image, this results in Sh1 positioning at the distal tip of the gonad.

The converse analysis with *inx-8(0) inx-9(0)* DTC and *inx-(+)* sheath cells is a bit more problematic. Although *inx-8(0) inx-9(0); tnEx202[lim-7p::inx-8::gfp]* hermaphrodites can be maintained as a self-fertile strain, they often produce short gonad arms (see Starich et al., 2014, Figure 15, C–E).

The authors now show that there is overlap in the DTC and Sh1 marked regions. Are these cells growing over one another or are the projections intercalated? Can this be shown by EM or super-resolution imaging?

Please see reply to Reviewer 1, Figure 1 supplements, and Videos 1–3. In short, for Class 2 gonads there can be “intercalation” when viewed as a maximum projection rendering. (However, as seen in 3D rendering, these cell processes can be quite far apart). When Sh1 is very far distal in a mutant genetic background (Video 3 example), we often observe apparent interference such that the DTC processes are located under Sh1. We think this is really interesting and deserves additional mechanistic study, but such an analysis is well beyond the scope of this manuscript.

Does the loss of innexin function in just Sh1 or the DTC change the border? This might help resolve their model about whether Sh1 is "sensing" reduced gap function number or function?

We thank the reviewer for this question, and we restated the point in our revision so as not to imply more than we intended: “sensed” was a poor word choice in retrospect. As shown in each “barbell” graph (green dots) and in the cognate box plots in relevant supplements, the position of the Sh1 border changes with respect to the distal end of the gonad. The position of the DTC is not significantly different in similarly marked strains with and without *inx-14(ag17)* (e.g., blue dots on “barbell” graphs and average DTC measurements in box plots) while the position of Sh1 is significantly different in each case. We therefore interpret the change as primarily a movement of Sh1 distally. As mentioned above, Starich et al., (2014) indicated that loss of *inx-8* and *inx-*9 from the Sh1 causes Sh1 to be mispositioned distally (see Figure 12A).

2) I would be reluctant to claim that they have "Consistent trends… from multiple markers." (line 194) Ideally, they would use different promoters expressed in the same tissue. If this cannot be done because other well characterized promoters do not exist, then the authors need to say, "Consistent results are obtained from independent transgenes"?

We have added to our analysis of the wild-type two markers with alternate Sh1-expressed promoters, as well as *inx-14(ag17)* with one of them. Notably, *fasn-1(av138[fasn-1::GFP])* is not a “transgene” as such. It is a CRISPR/Cas9 GFP insertion into the genomic locus. See revised Figure 1—figure supplement 1.

Similarly in the paragraph line 219 -226, it is unclear what you meant to test here is the strain was first tested with one of the transgenes.

We interpret the reviewer to be asking, “It is unclear what you meant to test here IF the strain was first tested with one of the transgenes”? In any case, here we are challenging the result presented in Figure 1—Figure Supplement 1 panel C of Gordon et al. (2020). We only see the result presented in Gordon et al. (2020) in 15% of gonads. We have re-written the text to be more explicit.

3) The authors argue that the qy78 fusion protein acts an antimorph and a loss-of-function to rigorously test these statements they should show the behaviour of qy78/+ and qy78/null. This would allow them to determine if it is a hypermorph, antimorph, or neomorph and if truly recessive.

We have generated and analyzed these strains, as well as going further by analyzing *inx-8(0)/+*. The results are in new Figure 2—figure supplement 2 panels B–E. In short, with respect to the distal Sh1 phenotype, *inx-8(qy78[mKate2::INX-8]* is dominant and dosage-dependent. That is, the Sh1 distal mis-positioning defect of q*y78/+* is similar to *qy78/qy78*, and the defect is partially mitigated in q*y78/0,* indicating that either the interaction of the fusion protein with the wild-type protein or the presence of the fusion protein in two copies causes the most penetrant defects, while insufficient quantity of the fusion protein is not as deleterious. We have added these results and a discussion of their implications.

4) qy78tn2031 appears to be slightly more abnormal than the tn2034 (Figure 2-Suppl 2B) It might be worth acknowledging that the slight difference could be due to strain background variation? Or are there other reasons for this difference?

Assuming the reviewer was actually referring to original Figure 2—figure supplement 1 (now revised supplement 2)? As shown in panel C of that figure (now panel E), with respect to mean process length and distal position of Sh1, these two strains are virtually identical. Panel B just shows examples. The length of the longest DTC process is indeed more variable in *qy78tn2031*.

5) Lines 252-254: From table 1, it does not appear that inx-14 suppresses qy78. Also, the interpretation that inx-9 suppresses is also complicated by the fact that the original qy78 alleles gave brood sizes of ~155 whereas the outcross to the DG strains have lower brood sizes. This needs to be clarified.

See above response to editor, with respect to Table 1 corrections, which should provide the requested clarification.

6) Line 326: Re "no significant change in the size of the SYGL-1-positive stem cell pool". The authors are not quantifying either the size of the stained region or (more appropriately) the number of nuclei that have SYGL-1 staining, just the extent of reach of the marker. If anything, a subset of the mutant germ lines (Figure 3) appear smaller than control. This should be addressed.

The use of the term “size” in line 326 was our editorial error which we regret, and which we have removed here and elsewhere. In other instances where we talk about the stem cell pool, the phrases “proximal border” and “proximal extent” were (and in revision, are) used. In the revised manuscript, we do not refer to “size”, but rather to the position of borders.

By way of further explanation, the relevant measurement here is the *proximal border* of the SYGL-1-positive cell pool relative to the *distal border* position of Sh1. This measure addresses the issue of whether Sh1 can orient the spindle of germ cells as they leave the stem cell pool, as claimed by Gordon et al. (2020). We made these measurements both in microns in main figures 3 and 4 and in “cell diameters” (CD) in supplements to figures 3 and 4. The reason the size/number of cells in the stem cell pool is not the relevant measure is that the thickness/thinness of the PZ can change the number of stem cells without altering the proximal border of the stem cell pool. An example is *daf-2(e1370)* which is thinner than wild type (Michaelson et al., 2010).

Given the importance of the measured position of the proximal border of the SYGL-1-positive zone/stem cell pool, we have added new Figure 3—figure supplement 2 and referred to it in the Materials & Methods. The Supplemental Figure is a graph of SYGL-1 level versus distal-proximal position, where the proximal extent of the SYGL-1-positive region as determined by image inspection, is compared with the SYGL-1 level measured by staining intensity, relative to the distal position of Sh1. The point here is that we reproducibly “call” the CD position for the SYGL-1 proximal border by eye, and that this position corresponds to a slightly greater than 90% fall in SYGL-1 level. In addition, the figure demonstrates that the SYGL-1 expression levels are very similar between genotypes, independent of the distal position of Sh1.

7) Similarly in this entire section, while it is clear that Sh1 is shifted distally (almost to the distal end of the germ line), it would be nice to know how the authors quantify proximal shift since the data is in microns which are not normalized for the total size of the germ line. Is there a change in the total number of nuclei or the numbers of rows?

This is why we measured in both µm (main Figures 3 and 4) and “cell diameter” (CD; supplements to Figures 3 and 4). Please also see explanation in previous point.

8) No mention is made in the text of Figure 4 qy78tn2031 or tn2034 or of Figure 4D or Figure 4 figure supplement 1.

We have amended the manuscript to refer to these.

9) Table 1 is very confusing because so much of the information in the footnotes refers to methods. It is also unclear, but does b represent the cumulative data from 2 different outcrosses both of which decreased brood size? Do they think there is a suppressor in the original strain? Either all the info in Table 1 footnotes should be included in a supplemental table or in the methods.Also, it might be helpful to consolidate a lot of the information into one large table-specifically added a column to Table 1 that includes a yes/no or distance for positional information of Sh1. (maybe "Sh1 edge moved distally")

We thank the reviewer for these comments. We agree. In revised Table 1, we included strain names, added an indication of Sh1 position phenotype on the table, and we removed the methods from the legend. The yes/no information regarding Sh1 was complicated by the fact that some strains could not be directly measured (due to the need for markers). See revised Table 1 legend.

10) The graphics of the genes and mutations should be combined into one figure.

We appreciate the reviewer’s point. However, we prefer to introduce the gene graphics in the same figure where they are relevant.

There were a number of textual edit that need to be made:Textual EditsLine 241: "must not appreciably perturb the function of inx-9" Please clarify why you think this might be the case?

We opted to delete this point as it was confusing.

Lines 833-834: This parenthetical about pSUN-1 pSer8 not being shown should be included in the figure legends since it is mentioned in the text as being used to define the border and is not shown in the figure. Or simply add "(see methods)" after the "pSUN-1(-) progenitor pool".

This is corrected.

Line 52: "Germ line" should be one word.

This is corrected.

Line 57: need a comma after "(DTC)".

This is corrected.

Line 62: Additional support for what?

We added “additional support to the germ line”. We also explain in the very next sentence.

Line 57 and 63: Figure 1C (just listed as Figure 1) is mentioned before Figures 1A and 1B.

This is corrected.

Figure 1A: white "inx-21" hard to read on the yellow background. Also, the proteins names should be capitalized not lower case.

This is corrected.

Line 80: "germline" should be "germ line".

This is corrected.

Line 81-83: The lof analysis does not follow from the rescue study. These should be separate sentences.

We modified the sentences.

Line 92: Should read "Figure 1C and D".

This is corrected.

Lines 155-156: the order of inx-8 and inx-9 needs to be reversed (written 9 then 8).

This is corrected.

Line 169: remove comma after: "2020))".

This is corrected.

Line 170-171: either remove comma before "as well as" or add comma after "inx-8 mutants".

We edited the text.

Line 216: The parenthetical is confusing as it comes after reference to inx-9 null allele, but refers to inx-8.

We edited the text.

Lines 223-225: the sentences are repeated.

We edited the text.

Line 226: do you have a reference for this (although common knowledge in the worm field it would be helpful). This is however integrated at the inx-8 locus, so it would be helpful to comment on and reference why this might be seen with single copy integrants.

We note that the issue is not with the single copy integrants. The wording is now changed to indicate that “both cases” refers to bcIs39 or tnIs5 which are both integrated arrays.

Line 244: Rather that "further" perhaps say "in addition" or "moreover". Also, the space between the alleles "qy78 tn2031" should be deleted. (also in Figure 2A).

We took the reviewer’s suggestion.

Line 250: I would say "adhesion functions" rather than "rivet functions" since the latter is an analogy to describe the adhesion role. Also "germline" should be "germ line" here and throughout since soma is a noun, presumably germ line should be as well.

We took the reviewer’s suggestion.

Lines 323-324: reference for the sygl-1(OLLAS) should be provided.

We took the reviewer’s suggestion.

Line 326: For consistence you should write "inx-8(qy78[mKate2::inx-8])" throughout or "inx-8(qy78)" but not both. Personally, the former is more helpful is reading the text.

With the exception of the somewhat complicated discussion of dosage we opted for *inx-8(qy78[mKate2::inx-8])* throughout.

Line 333: "each other" should be "one another".

We took the reviewer’s suggestion.

Line 343: "1-2 cell diameters". What is the basis of this claim?

We added additional text to the introduction and to the results that we hope clarifies this point.

Line 439: Add a comma after "organ system".

We took the reviewer’s suggestion.

Line 495: is there a funding stream for Wormbase that can be cited?

We are following instructions on WormBase

https://wormbase.org/about/citing_wormbase#0123--10

which state: “Include in your acknowledgments, a statement thanking WormBase, ‘We thank WormBase’. To save characters, it can be included in a list: ‘We thank XYZ for reagents, ABC for comments and WormBase’. If this is a problem, just mention the particular WormBase release (WSnnn) in the text. When you use these methods, the acknowledgements of WormBase are amenable to searching and reporting.”

Lines 686-687: I am presuming the chromosome insertion sites for cpIs122 and naIs37 are not known?

In the course of revisions, we mapped *naIs37* to the right arm of I. We have corrected genotypes to indicate this. We did not find anything in Linden et al. 2017 regarding insertion site for *cpIs122*.

Line 832: "until" Not "till".

We took the reviewer’s suggestion.

Reviewer#3 (Recommendations for the authors):Explanation of Figure 2 is a bit confusing. When they used CRISPR approach to remove poisonous inx-8-mKate gene, is the animal examined homozygous for inx-8 deletion, or heterozygous over wild type allele? If the claim of inx-8-mKate being poisonous is correct, inx-8-mKate/+ would also have phenotypes (reduced brood size, loss of the bare region), and the removal of mKate allele (in the presence of inx-8 wild type allele) should rescue the phenotype entirely. Alternatively, if the authors mean that inx-8-mkate's phenotype can be only observed as homozygote, it might mean that inx-8-mKate is a loss of function allele. Also, I see the possibility that the authors imply that inx8-mKate is poisonous to inx-9, and poisoning effect can be seen only as homozygote? Whereas the data itself is convincing, the terminology (poisonous = dominant-negative, hypomorph) as well as the description of genetics (heterozygote vs. homozygote) that can guide the determination of poisonous vs. hypomorph, is unclear to the readers. This should be clarified.

We have amended the text to address this point.

[Editors' note: further revisions were suggested prior to acceptance, as described below.]

Reviewer #1 (Recommendations for the authors):In the revised version of the manuscript, Tolkin et al. addressed all of reviewers' suggestions and included new data that significantly strengthened the manuscript. Especially notable are: 1) the inclusion of the new classification of DTC/Sh1 process positioning. It is very helpful to address morphological variation and to allow clear comparison among different strains and markers of DTC and Sh1. 2) the inclusion of immunostaining with anti-INX-8. This clearly removes the criticism of sub-optimal markers or a concern that distinct Sh1 boundary locations are attributable to using different markers. Anti-INX-8 clearly shows distinct locations of Sh1 boundary in the wild type vs inx-8(qy78[mKate::inx-8]) backgrounds.Overall, this work challenges the Gordon et al., 2020 model of oriented cell divisions in C. elegans germline and presents a significant advance in the field.

We thank the reviewer for the positive assessment and very much appreciate their critiques of the original submission, as well as the revised manuscript.

Reviewer #2 (Recommendations for the authors):The revised manuscript addresses all of the major concerns from the previous submission by using multiple reporters for each cell type, by creating a classification system, by providing EM reconstruction of the DTC/SH1 cell and processes, and by redoing all of the viability assays to address strain differences between labs. The importance of the paper remains both in its tale of woe for transgenic reporters, but more importantly, by providing a framework for detailed characterization and classification of cell:cell interactions. It also raises new and interesting questions about the roles of the sheath cells and that innexin proteins might slight haploinsufficiency, a phenotype that may be relevant to human disease.

We thank the reviewer for their positive comments and appreciate the thought, care, and attention the reviewer gave to both the original submission and the revised manuscript. These comments were very helpful in improving the quality of the work and the clarity of the presentation.

There are minor changes to the text that should be addressedLine 138- 139. Technically the images in Figures 1 and suppl do not show the germ cells proper. Move the comment about DIC imaging from Line 163 to line 138 for clarification.

We thank the reviewer. The text from line 138 was meant to introduce two points: the observed variability of Sh1 position, and what characterizes “bare regions”, thereby setting up a series of observations and measurements, including visualization of germ cells by DIC microscopy, that further substantiate the existence of the bare regions in the gonad. We felt it necessary to explain first the system for classification and the 8 µm threshold in the subsequent paragraphs before stating that we confirmed with DIC. Therefore, we modified these introductory sentences slightly as follows, and put the references to specific figures in the relevant subsequent sections.

The text now reads,

“As detailed below, we found that Sh1 is somewhat variably positioned in wild-type young adult hermaphrodites. We confirmed the existence of bare regions between Sh1 and the DTC, characterized by germ cells that do not contact Sh1 or the DTC.”

Lines 261-267: The change in the allele names here is confusing and would help it were described in the text itself rather than just the Materials and methods.

We revised the text to explain why the *mKate2::inx-8* allele name changes. The revised text (lines 278–286) now reads, “The same mKate2::INX-8 fusion, independently generated in an *inx-9(ok1502)* null mutant background (i.e., *inx-8(qy102[mKate2::inx-8])*, see Materials and methods, Gordon et al., 2020), was also observed to shift Sh1distally in *inx-8(qy102[mKate2::inx-8]) inx-9(ok1502)* double mutants. In contrast, loss of *inx-9* alone does not significantly affect Sh1 position (Figure 2–—figure supplement 2). If mKate::INX-8 were innocuous, the *inx-8(qy102[mKate2::inx-8)]* allele in the absence of *inx-9* (*inx-9(1502)*) should have had no effect on Sh1 position.”

There are two odd source file to PDF conversion issues that you should check carefully is accepted for publication:– In all figures in the PDF, the gene names and alleles are not italicized in all figures and graphs. But in the source files, they seemed correct (italic)!– In Figure 1B and 2A, the text of the alleles appears to be shadowed oddly (in the PDF- but not the source files). Please confirm there is not an issue with the text in that figure. Also in Figure 1 Suppl 1A, the pink text.

We thank the reviewer for pointing out some file-conversion issues that were a consequence of converting image files to pdf files to generate the files for the review process. All of the original source figure files look ok, as mentioned by the reviewer. We have mentioned this issue in our cover letter, and we hope that should our manuscript be accepted, the figures would lack these file conversion effects.

p. 3 line 1 add comma after interactions.

Done, thanks!

Line 83-84 need a reference. Is this the paper you are refuting or other? It seems you are saying that Sh1 drives cell fate determining germ cell divisions, since you said that ectopic inx-8 in the Sh1 in inx8(0) inx-9(0) rescued proliferation. Isn't this contradicting what you are setting out to prove?

We thank the reviewer for helping us tighten up the text in this portion of the manuscript. We have provided the reference, which is Starich et al., 2014. We have also edited the text for enhanced precision to read,

“Restoration of *inx-8* either to the DTC or to sheath cells rescues the severe germline proliferation defect of the *inx-8(0) inx-9(0)* double mutant (Starich et al., 2014).”

This introductory information is provided to set the stage for the analysis and in no way contradicts what we are setting out to examine in the manuscript. The innexins are absolutely essential for germ cells to proliferate and the double null mutant gives as severe an effect on germ cell number as null mutations in *glp-1/Notch*. Moreover, the lack of proliferation in *inx-8(0) inx-9(0)* double null mutants is “epistatic” to a dominant gain-of-function Tumorous allele of *glp-1/Notch*. One interpretation is that small molecules that transit through these junctions are provided from the soma to the germline to enable germ cell proliferation.

To point out that the innexin mutant alleles used in this study are able to promote robust proliferation of germ cells but nevertheless affect the position of Sh1, we have added the sentence, “The mutant innexin allele combinations used in this study are proficient in promoting germline proliferation; however, as this work shows, many of them affect the position of Sh1 with respect to the distal end of the gonad.” This change provides a nice transition into the next paragraph, which discusses the role of Sh1 in gonadal architecture.

Line 138 appears to be missing words or punctuation.

Thank you for pointing this out. We added the missing word “between” and made minor edits as noted above.

Line 168, "wild-type unmarked young adult" add come after "wild-type".

Done, thanks!

Kerning of text in Figure 2 Suppl 1 and Suppl 2B above the Figuresis off (again in my PDF, but is this true for all?)

Thanks! We have checked the figure files and are sure these appear to be file conversion artefacts, as noted above.

Line 285 "…in one but not both cells.." should read "in one, but not both, cells".

Fixed, thanks!

Line 320, the statement "that are lacking in inx-8(qy78[mKate2::inx-8])" is confusing and makes it seem that you are saying it is lacking in mKate (which it is due to the deletion) not that it dissimilar from qy78 which I what I think you intended. I would just remove this part of the sentence.

We rewrote the two sentences for increased clarity. They now read,

“Consistent with our hypothesis, the deletion mutant *inx-8(qy78tn2031)* displays a significantly lower frequency of Class 3 gonads than the original *inx-8(qy78[mKate2::inx-8])* allele. Sh1 in the deletion mutant is positioned more proximally, and therefore, like the wild type, exhibits bare regions that are lacking in *inx-8(qy78[mKate2::inx-8])* mutants (Figure 2D–F).”

Line 381 need reference to a figure.

Thank you! We included a reference to Figure 3C.

Line 385, italicize "per se".

Done.

Line 499, italicize "Xenopus".

Done.

Line 819 "Embryonic" should be lower case.

Fixed, thanks!

There should be no hyphen for "P-value". And the "P" should be italicized "P".

Fixed, thanks!

Reviewer #3 (Recommendations for the authors):This manuscript by Tolkin et al. provide the evidence that a previously-published eLife paper by Gordon et al. was misled by the use of inx-8-GFP, which Tolkin found to function as a dominant negative form. Accordingly, this manuscript substantially changes our current understanding of how germline stem cells and their differentiation are regulated in the C. elegans gonad.

We thank the reviewer for the positive comments on our work and its importance.

Whereas there remain some unresolved discrepancies between this manuscript and Gordon et al., the revised version of Tolkin et al. manuscript strongly supports their main conclusions, including the presence of 'bare region' (the area of germ cells that are not touched by Sheath cells or Distal Tip Cells), and the germ cell biology supported by such an architecture of the gonad. Even if some discrepancies may end up to be corrected in the future, their main conclusion is firm enough that merits publication in eLife.

Again, we thank the reviewer.

Whereas their conclusion 'Sh1 position is neither relevant for GLP-1/Notch signaling nor for the exit of germ cells from the stem cell pool' (in the abstract) is correct in that the junction of DTC and Sh1 cells do not serve as the interface for GSCs' 'asymmetric division' (as Gordon 2020 paper suggested), it seems that inx influences the position of Sh1 and also impacts brood size (maybe they think 250 vs. 210 are not different, but have they run statistics on these numbers? It looks to me that there are correlation between Sh1 position and a slight reduction in brood size?). If there is any impact of inx8 mutant/dominant negative, it should be a bit more carefully described. I recommend revising the text to more accurately describe their important conclusions (that innexin mutant reduces bare region with likely functional consequences).

The reviewer’s comment brings up an interesting question, “what is the biological significance of the bare regions?” The reviewer notes that innexin mutations that reduce or eliminate the bare regions often exhibit a modestly decreased brood size. However, we are unable to ascribe this effect to an absence of the bare regions *per se* because the innexins have multiple functions in germline development (Starich et al., 2014, 2020; Starich and Greenstein, 2020). We have addressed this point by revising the Discussion. We added the following text in a brief new paragraph as follows (lines 508–514),

“Our observation that loss of the bare regions correlates with only a relatively modest effect on brood size (Table 1) raises the question of what, if any, biological significance the bare regions might have. Because the innexins have multiple functions in germline development (Starich et al., 2014, 2020; Starich and Greenstein, 2020), we are unable to ascribe the observed brood-size reduction in strains lacking bare regions to the absence of the bare regions *per se*. Thus, any potential roles for the bare regions vis-à-vis brood size and germline development will require additional study.”